# Towards Unsupervised Content Disentanglement in Sentence Representations via Syntactic Roles

## Abstract

Linking neural representations to linguistic factors is crucial in order to build and analyze NLP models interpretable by humans. Among these factors, syntactic roles (e.g. subjects, direct objects,...) and their realizations are essential markers since they can be understood as a decomposition of predicative structures and thus the meaning of sentences. Starting from a deep probabilistic generative model with attention, we measure the interaction between latent variables and realizations of syntactic roles and show that it is possible to obtain, without supervision, representations of sentences where different syntactic roles correspond to clearly identified different latent variables. The probabilistic model we propose is an Attention-Driven Variational Autoencoder (ADVAE). Drawing inspiration from Transformer-based machine translation models, ADVAEs enable the analysis of the interactions between latent variables and input tokens through attention. We also develop an evaluation protocol to measure disentanglement with regard to the realizations of syntactic roles. This protocol is based on attention maxima for the encoder and on latent variable perturbations for the decoder. Our experiments on raw English text from the SNLI dataset show that *i)* disentanglement of syntactic roles can be induced without supervision, *ii)* ADVAE separates syntactic roles better than classical sequence VAEs and Transformer VAEs, *iii)* realizations of syntactic roles can be separately modified in sentences by mere intervention on the associated latent variables. Our work constitutes a first step towards unsupervised controllable content generation. The code for our work is publicly available[1].

## 1 Introduction

A disentangled representation of data describes information as a combination of separate *understandable* factors. This separation provides better transparency, but also better transfer performance (Higgins et al., 2018; Dittadi et al., 2021). When it comes to disentanglement, Variational Autoencoders (VAEs; Kingma & Welling, 2014) were extensively proven effective (Higgins et al., 2017; Chen et al., 2018; Rolinek et al., 2019). and were used throughout several recent works (Chen et al., 2019; Li et al., 2020b; John et al., 2020). In NLP, disentanglement has been mostly performed to separate the semantics (or content) in a sentence from characteristics such as style and structure in order to generate paraphrases (Chen et al., 2019; John et al., 2020; Bao et al., 2020; Huang & Chang, 2021; Huang et al., 2021). We show in our work that the information in the content itself can be separated with a VAE-based model. In contrast to the aforementioned works, we use neither supervision nor input syntactic information for this separation. We demonstrate this ability by controlling the lexical realization of core syntactic roles. For example, the subject in a sentence can be encoded separately and controlled to generate the same sentence with another subject. Our framework includes a model and an evaluation protocol aimed at measuring the disentanglement of syntactic roles.

The model we introduce is an Attention-Driven VAE (ADVAE), which we train on the SNLI raw text dataset (Schmidt et al., 2020). It draws its inspiration from attention-based machine translation models (Bahdanau et al., 2015; Luong et al., 2015). Such models translate sentences between languages with

---

[1]URL to be disclosed upon publication. Our code is provided as supplemental material during the submission process.

different underlying structures and can be inspected to show a coherent alignment between spans from both languages. Our ADVAE uses Transformers (Vaswani et al., 2017), an attention-based architecture, to map sentences from a language to independent latent variables, then map these variables back to the same sentences. Although ADVAE could be used to study other attributes, we motivate it (§4.1) and therefore study it for the alignment of syntactic roles with latent variables.

Evaluating disentanglement with regard to spans is challenging. After training the model and only for evaluation, we use linguistic information (from an off-the-shelf dependency parser) to first extract syntactic roles from sentences, and then study their relation to latent variables. To study this relation on the ADVAE decoder, we repeatedly *i)* generate a sentence from a sampled latent vector *ii)* perturb this latent vector at a specific location *iii)* generate a sentence from this new vector and observe the difference. On the encoder side, we study the attention values to see whether each latent variable is focused on a particular syntactic role in input sentences. The latter procedure is only possible through the way our ADVAE uses attention to produce latent variables. To the best of our knowledge, we are the first to use this transparency mechanism to obtain quantitative results for a latent variable model.

We first justify our focus on syntactic roles in §3, then we go over our contribution, which is threefold: *i)* We introduce the ADVAE, a model that is designed for *unsupervised* disentanglement of syntactic roles, and that enables analyzing the interaction between latent variables and observations through the values of attention (§4), *ii)* We design an experimental protocol for the challenging assessment of disentanglement over realizations of syntactic roles, based on perturbations on the decoder side and attention on the encoder side (§5), *iii)* Our empirical results show that our architecture disentangles syntactic roles better than standard sequence VAEs and Transformer VAEs and that it is capable of controlling realizations of syntactic roles separately during generation (§6).

## 2   RELATED WORKS

**Linguistic information in neural models**   Accounting for linguistic information provided better inductive bias in the design of neural NLP systems during recent years. For instance, successful attempts at capturing linguistic information with neural models helped improve grammar induction (RNNG; Dyer et al., 2016), constituency parsing and language modeling (ON-LSTM; Shen et al., 2019, ONLSTM-SYD; Du et al., 2020), as well as controlled generation (SIVAE; Zhang et al., 2019). Many ensuing works have also dived into the linguistic capabilities of the resulting models, the types of linguistic annotations that emerge best in them, and syntactic error analyses (Hu et al., 2020; Kodner & Gupta, 2020; Marvin & Linzen, 2020; Kulmizev et al., 2020). Based on the Transformer architecture, the self-supervised model BERT (Devlin et al., 2019) has also been subject to studies showing that the linguistic information it captures is organized among its layers in a way remarkably close to the way a classical NLP pipeline works (Tenney et al., 2020). Furthermore, (Clark et al., 2019), showed that many attention heads in BERT specialize in dependency parsing. We refer the reader to (Rogers et al., 2020) for an extensive review of Bert-related studies. However, such studies most often rely on structural probes (Jawahar et al., 2019; Liu et al., 2019; Hewitt & Manning, 2019) to explain representations, probes which are not without issues, as shown by Pimentel et al. (2020). In that regard, the generative capabilities and the attention mechanism of our model offer an alternative to probing: analysis is performed directly on sentences generated by the model and on internal attention values.

**Disentanglement in NLP**   The main line of work in this area revolves around using multitask learning to separate concepts in neural representations (*e.g.* style vs content (John et al., 2020), syntax vs semantics (Chen et al., 2019; Bao et al., 2020)). Alternatively, Huang & Chang (2021) and Huang et al. (2021) use syntactic trees *as inputs* to separate syntax from semantics, and generate paraphrases without a paraphrase corpus. Towards less supervision, Cheng et al. (2020) only uses style information to separate style from content in representations. Literature on *unsupervised* disentanglement in NLP remains sparse. Examples are the work of Xu et al. (2020) on categorical labels (sentiment and topic), and that of Behjati & Henderson (2021) on representing morphemes using character-level Seq2Seq models. The work of Behjati & Henderson (2021) is closest to ours as it uses Slot Attention (Locatello et al., 2020), which, like ADVAE, is a cross-attention-based representation technique. Our contribution depart from previous work since *i)* syntactic parses are not used as learning signals but as a way to interpret our model, and *ii)* cross-attention enables our model to link a fixed number of latent variables to text spans.

## 3 SYNTACTIC ROLES AND DEPENDENCY PARSING

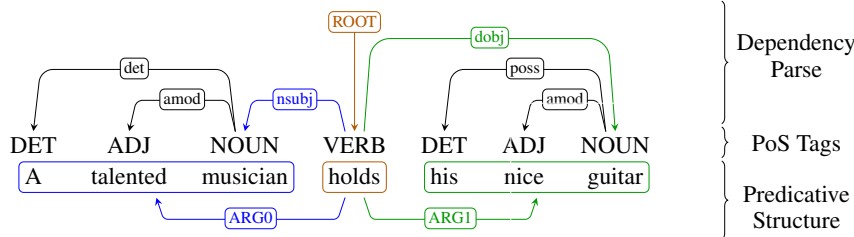

Figure 1: A sentence and its syntactic roles. The correspondence between syntactic roles and elements of the predicative structure is highlighted with colors.

We present in Figure 1 an example sentence with its dependency parse[2], its Part-of-Speech (PoS) tags, and a flat predicative structure with PropBank-like semantic roles (Palmer et al., 2005). Dependency parsing yields a tree, where edges are labeled with syntactic roles (or relations or functions) such as *nominal subject* (*nsubj*). The lexical realizations of these syntactic functions are textual spans and correspond to syntactic constituents. For instance, the lexical realization of the *direct object* (*dobj*) of the verb *holds* in this sentence is the span *his nice guitar*, with *guitar* as head. In short, the spans corresponding to subtrees consist of tokens that are more dependent of each other than of the rest of the sentence. As a consequence, and because a disentanglement model seeks independent substructures in the data, we expect such a model to converge to representations that display separation in realizations of frequent syntactic roles.

In our work, we focus[3] within the same framework. on nominal subjects, verbal roots of sentences, and direct or prepositional objects. These are *core* (as opposed to *oblique*; see Nivre et al. (2016) for details on the distinction) syntactic roles, since they directly relate to the predicative structure. In fact in most cases, as illustrated in Figure 1, the verbal root of a sentence is its main predicate, the nominal subject its agent (*ARG0*) and the direct or prepositional object its patient (*ARG1*).

## 4 MODEL DESCRIPTION

The usual method to obtain sentence representations from Transformer models uses only a Transformer encoder either by taking an average of the token representations or by using the representation of a special token (*e.g* [CLS] in BERT(Devlin et al., 2019)). Recently, the usage of both Transformer encoders and decoders has also been explored in order to obtain representations whether by designing classical Autoencoders (Lewis et al., 2019; Siddhant et al., 2019; Raffel et al., 2020), or VAEs (Li et al., 2020a). Our model, the ADVAE, differs from these models in that it uses *targets* in Machine Translation (MT) Transformers (*i.e* an encoder and a decoder) to produce sentence representations. Producing representations with Cross-Attention has been introduced by Locatello et al. (2020) as part of the Slot Attention modules in the context of unsupervised object discovery. However, in contrast to Locatello et al. (2020), we simply use Cross-Attention as it is found in Vaswani et al. (2017), *i.e.* without normalizing attention weights over the query axis, or using GRUs(Cho et al., 2014) to update representations. As will be shown through our experiments, this is sufficient to disentangle syntactic roles. We explain the observation that motivates our work in §4.1, we then describe in §4.2 the minimal changes we apply to MT Transformers, and finally, we present the objective we use in §4.3. The parallel between our model and MT Transformers is illustrated in Figure 2.

### 4.1 THE INTUITION BEHIND OUR MODEL

Consider $s = (s_j)_{1 \leq j \leq N_s}$ and $t = (t_j)_{1 \leq j \leq N_t}$, two series of tokens forming respectively a sentence in a source language and a sentence in a target language. Given $s$, attention-based translation models

---

[2]Following the ClearNLP constituent to dependency conversion, close to Stanford Dependencies de Marneffe & Manning (2008). See `https://github.com/clir/clearnlp-guidelines/blob/master/md/components/dependency_conversion.md`.

[3]Future research that takes interest in the finer-grained disentanglement of content may simply study a larger array of syntactic roles. Using our current system we display results including all syntactic roles in Appendix G.

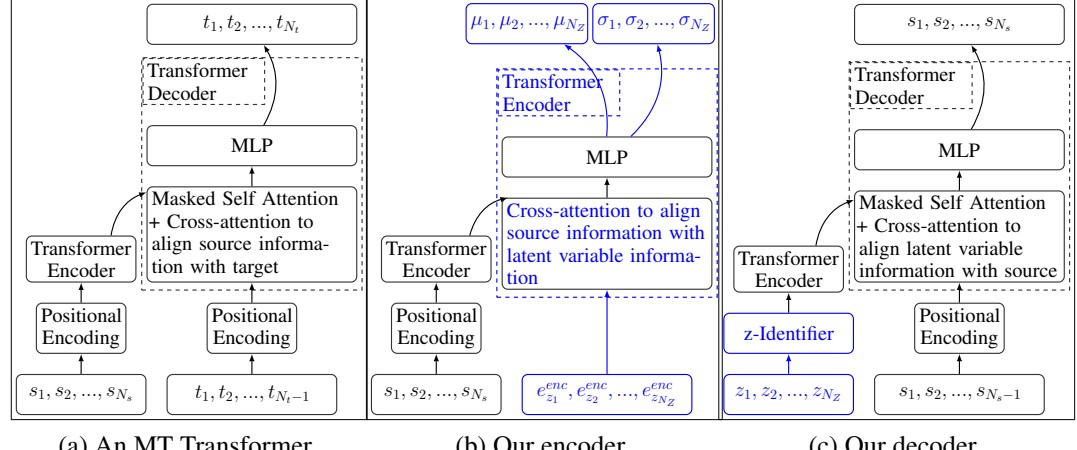

|  (a) An MT Transformer | (b) Our encoder | (c) Our decoder |

Figure 2: In blue, we highlight in (b) the difference between our encoder and a source-to-target MT model, and in (c) the difference between our decoder and a target-to-source MT model. The input at the bottom right for the Transformer Decoders in (a) and (c) is the series of previous words for autoregressive generation. The input to our model is a series of words $s$, at the bottom left of (b), and its output is the reconstruction of these words in the same language, at the top right of (c).

are capable of yielding $t$ while also providing information about the alignment between the groups of tokens (of different sizes) in both sentences (Bahdanau et al., 2015; Luong et al., 2015)). This evidence suggests that attention-based architectures are capable of factoring information from groups of words according to a source structure, and redistributing it according to a target structure.

The aim of our design is to use, as a target, a set of $N_Z$ *independent* latent variables that will act as fixed placeholders for the information in sentences. We stress that $N_Z$ is fixed and independent of the input sentence size $N_s$. Combining Transformers, an attention-based MT model, and the VAE framework for disentanglement, our ADVAE is intended to factor information from independent groups of words into separate latent variables. In the following sections, we will refer to this set of independent latent variables as the latent *vector* $z = (z_i)_{1 \leq i \leq N_Z}$ and to each $z_i$ as a latent *variable*.

## 4.2 MODEL ARCHITECTURE

**Inference model:** This is the inference model $q_\phi$ (encoder in Fig. 2.b) for our latent variables $z = (z_i)_{1 \leq i \leq N_Z}$. It differs from an MT Transformer in two ways. First it uses as input a sentence $s$, and $N_Z$ learnable vectors $(e_{z_i}^{enc})_{1 \leq i \leq N_Z}$ instead of the target tokens $t$ used in translation. These learnable vectors will go through Cross-Attention without Self-Attention. We stress that these learnable vectors are input-independent. Second its output is not used to select a token from a vocabulary but rather passed to a linear layer (resp. a linear layer followed by a softplus non-linearity) to yield the mean parameters $(\mu_i)_{1 \leq i \leq N_Z}$ (resp. the standard deviation parameters $(\sigma_i)_{1 \leq i \leq N_Z}$) to parameterize the diagonal Gaussian distributions $(q_\phi^{(i)}(z_i|s))_{1 \leq i \leq N_Z}$. The Transformer Decoder is therefore replaced in Fig 2.b by a Transformer Encoder that uses Cross-attention to factor information from the sentence. The distribution of the whole latent vector is simply the product of Gaussians $q_\phi(z_1, \ldots, z_{N_Z}|s) = \prod_i^{N_Z} q_\phi^{(i)}(z_i|s)$.

**Generation model:** Our generation model consists of an autoregressive decoder (Fig. 2.c) $p_\theta(s|z_1, \ldots, z_{N_Z}) = \prod_j^{N_s} p_\theta(s_j|s_{<j}, z_1, \ldots, z_{N_Z})$ where $s_{<i}$ is the series of tokens preceding $s_i$, and a prior assuming independent standard Gaussian variables, *i.e.* $p(z_1, \ldots, z_{N_Z}) = \prod_i^{N_Z} p(z_i)$.

Each latent variable $z_i$ is concatenated with an associated learnable vector $e_{z_i}^{dec}$ (*z-Identifier* in Fig. 2.c) instead of going through positional encoding. From there on, the latent variables are used like source tokens in an MT Transformer.

### 4.3 OPTIMIZATION OBJECTIVE

We train our ADVAE using the $\beta$-VAE (Higgins et al., 2017) objective, which is the Evidence Lower-Bound (ELBo) with a controllable weight on its Kullback-Leibler (KL) term:

$$\log p_\theta(s) \geq \mathbb{E}_{(z) \sim q_\phi(z|s)} \left[ \log p_\theta(s|z) \right] - \beta D_{\mathrm{KL}}[q_\phi(z|s)||p(z)] \tag{1}$$

In Eq. 1, $s$ is a sample from our dataset, $z$ is our latent vector and the distributions $p_\theta(s) = \int p_\theta(s|z)p(z)dz$ and $q_\phi(z|s)$ are respectively the generation model and the inference model. We use a standard Gaussian distribution as prior $p(z)$ and a diagonal Gaussian distribution as the approximate inference distribution $q_\phi(z|s)$. The weight $\beta$ is used (as in Chen et al., 2018, Xu et al., 2020, Li et al., 2020b) to control disentanglement, but also to find a balance between the expressiveness of latent variables and the generation quality.

## 5 EVALUATION PROTOCOL

In order to quantify disentanglement, we first measure the interaction between latent variables and syntactic roles. To do so, we extract *core* syntactic roles from sentences according to the procedure we describe in §5.1. Subsequently, for the ADVAE decoder, we repeatedly perturb latent variables and measure their influence on the realizations of the syntactic roles in generated sentences (§5.2). For the ADVAE encoder, we use attention to determine the syntactic role that participates most in producing the value of each latent variable (§5.3).

Given these metrics, we measure disentanglement taking inspiration from the Mutual Information Gap (MIG; Chen et al., 2018) in §5.4. MIG consists in measuring the difference between the first and second latent variables with the highest mutual information with regard to a target factor. It is intended to quantify the extent to which a target factor is concentrated in a single variable. This metric assumes knowledge of the underlying distribution of the target information in the dataset.However, there is no straightforward or agreed-upon way to set this distribution for text spans, and therefore to calculate MIG in our case. As a workaround, we use the influence metrics defined in §5.2 and §5.3 as a replacement for mutual information to quantify disentanglement.

### 5.1 SYNTACTIC ROLE EXTRACTION

We use the Spacy[4] dependency parser (Honnibal & Montani, 2017) trained on Ontonotes5 (Weischedel et al., 2013). For each sentence the realization of *verb* is the root of the dependency tree if its POS tag is *VERB*. Realizations of *subj* (subject), *dobj* (direct object), and *pobj* (prepositional object) are *spans* of subtrees whose roots are labelled resp. *nsubj*, *dobj*, and *pobj*.

In the rare cases where multiple spans answer the requirement for a syntactic role, we take the first one as the subsequent spans are most often part of a subordinate clause. A realization of a syntactic role in $R = \{verb, subj, dobj, pobj\}$ is empty if no node in the dependency tree satisfies its extraction condition.[5]

### 5.2 LATENT VARIABLE INFLUENCE ON DECODER

Intuitively, we repeatedly compare the text generated from a sampled latent vector to the text generated using the same *vector* where only one latent *variable* is resampled. Thus we can isolate the effect of each latent *variable* on output text and gather statistics.

More precisely, we sample $T^{dec}$ latent *vectors* $(z^{(l)})_{1 \leq l \leq T^{dec}} = (z_i^{(l)})_{1 \leq l \leq T^{dec}, 1 \leq i \leq N_Z}$. Then for each $z^l$, and for each $i$ we create an altered version $\tilde{z}^{(li)} = (\tilde{z}_{i'}^{(li)})_{1 \leq i' \leq N_Z}$ where we resample only the $i$th latent *variable* (i.e. $\forall i' \neq i$, $\tilde{z}_{i'}^{(li)} = z_{i'}^{(l)}$).

Generating the corresponding sentences[6] with $p_\theta(s|z)$ yields a list of original sentences $(s^{(l)})_{1 \leq l \leq T^{dec}}$, and a matrix of sentences displaying the effect of modifying each latent variable

---

[4]https://spacy.io/models/en#en_core_web_sm
[5]Examples of syntactic role extractions can be found in Appendix D.
[6]Throughout this work, we use greedy sampling (sampling the maximum-probability word at each step), for all generated sentences.

$(\tilde{s}^{(li)})_{1 \leq l \leq T^{dec}, 1 \leq i \leq N_Z}$. For each syntactic role $r \in R$, we will denote the realization extracted from a sentence $s$ with $\rho_r(s)$.

To measure the influence of a variable $z_i$ on the realization of a syntactic role $r$, denoted $\Gamma_{ri}^{dec}$, we estimate the probability that a change in this latent variable incurs a change in the span corresponding to the syntactic role. We first discard, for the influence on a role $r$, sentence pairs $(s^{(l)}, \tilde{s}^{(li)})$ where it appears or disappears, because the presence of a syntactic role is a property of its parent word, (*e.g.* the presence or absence of a *dobj* is controlled by the *transitivity* of the verb) hence not directly connected to the representation of the role $r$ itself. As they are out of the scope of our work, we report measures of these structural changes (diathesis) in Appendix C, and leave their extensive study to future works. We denote the remaining number of samples $T_{ri}^{\prime dec}$.

In the following, we use operator $\mathbf{1}\{.\}$, which is equal to 1 when the boolean expression it contains is true and to 0 when it is false. This process yields a matrix $\Gamma^{dec}$ of shape $(|R|, N_Z)$ which summarizes interactions in the *decoder* between syntactic roles and latent variables:

$$\Gamma_{ri}^{dec} = \sum_{l=1}^{T_{ri}^{\prime dec}} \frac{\mathbf{1}\{\rho_r(s^{(l)}) \neq \rho_r(\tilde{s}^{(li)})\}}{T_{ri}^{\prime dec}} \qquad (2)$$

## 5.3 ENCODER INFLUENCE ON LATENT VARIABLES

We compute this on a held out set of size $T^{enc}$ of sentences $(s_j^{(l)})_{1 \leq l \leq T^{enc}, 1 \leq j \leq N_{s(l)}}$. Each sentence $s^{(l)}$ of size $N_{s(l)}$ generates an attention matrix $(a_{ij}^{(l)})_{1 \leq i \leq N_Z, 1 \leq j \leq N_{s(l)}}$. Attention values are available in the Transformer Encoder with cross-attention computing the inference model[7], and quantify the degree to which each latent variable embedding $e_{z_i}^{enc}$ draws information from each token $s_j$ to form the value of $z_i$.

For the encoder, we consider the influence of a syntactic role on a latent variable to be the probability for the attention values of the latent variable to reach their maximum on the index of a token in that syntactic role's realization. The indices of tokens belonging to a syntactic role $r$ in a sentence $s^{(l)}$ are denoted $\arg_r(s^{(l)})$. For each syntactic role $r$ and sentence $s^{(l)}$, we discard inputs where this syntactic role cannot be found, and denote the remaining number of samples $T_r^{\prime enc}$. The resulting measure of influence of syntactic role $r$ on variable $z_i$ is denoted $\Gamma_{ri}^{enc}$. The whole process yields matrix $\Gamma^{enc}$ of shape $(|R|, N_Z)$ which summarizes interactions in the *encoder* between syntactic roles and latent variables:

$$\Gamma_{ri}^{enc} = \sum_{l=1}^{T_r^{\prime enc}} \frac{\mathbf{1}\{\arg\max_j(a_{ij}^{(l)}) \in \arg_r(s^{(l)})\}}{T_r^{\prime enc}} \qquad (3)$$

## 5.4 DISENTANGLEMENT METRICS

For $\Gamma^*$ (either $\Gamma^{dec}$ or $\Gamma^{enc}$) each line corresponds to a syntactic role in the data. The disentanglement metric for role $r$ is the following:

$$\Delta\Gamma_r^* = \Gamma_{rm_1}^* - \Gamma_{rm_2}^* \qquad (4)$$
$$s.t. \quad m_1 = \arg\max_{1 \leq i \leq N_Z} \Gamma_{ri}^*, \quad m_2 = \arg\max_{1 \leq i \leq N_Z, j \neq m1} \Gamma_{ri}^*$$

We calculate total disentanglement scores for syntactic roles using $\Gamma^{dec}, \Gamma^{enc}$ as follows:

$$\mathbb{D}_{dec} = \sum_{r \in R} \Delta\Gamma_r^{enc}, \quad \mathbb{D}_{enc} = \sum_{r \in R} \Delta\Gamma_r^{enc} \qquad (5)$$

In summary, the more each syntactic role's information is concentrated in a single variable, the higher the values of $\mathbb{D}_{dec}$ and $\mathbb{D}_{enc}$. However, similar to MIG, these metrics do not say whether variables

---

[7]For simplicity, attention values are averaged over attention heads and transformer layers. This also allows drawing conclusions with regard to the tendency of the whole attention network, and not just particular specialized heads as was done in Clark et al. (2019). Nevertheless, we display per-layer results in Appendix J.

capturing our concepts of interest are *distinct*. Therefore, we also report the number of distinct variables that capture the most each syntactic role (*i.e* the number of distinct values of $m_1$ in Eq. 4 when looping over $r$). This is referred to as $N_{\Gamma^{enc}}$ for the encoder and $N_{\Gamma^{dec}}$ for the decoder.

## 6 EXPERIMENTS

**Dataset** Previous unsupervised disentanglement works (Higgins et al., 2017; Kim & Mnih, 2018; Li et al., 2020b) tend to use relatively homogeneous and low complexity data. The data has *low complexity* if it varies along clear factors which correspond to what the model aims to disentangle. Similarly, we use a dataset where samples exhibit low variance in terms of syntactic structure while providing a high diversity of realizations for the syntactic roles composing the sentences, which is an adequate test-bed for unsupervised disentanglement of syntactic roles' realizations. This dataset is the plain text from the SNLI dataset (Bowman et al., 2015) extracted[8] by Schmidt et al. (2020). The SNLI data is a collection of premises (on average $8.92 \pm 2.66$ tokens long) made for Natural Language Inference. We use 90K samples as a training set, 5K for development, and 5K as a test set.

**Setup** Our objective is to check whether the architecture of our ADVAE induces better syntactic role disentanglement. We compare it to standard Sequence VAEs (Bowman et al., 2016) and to a Transformer-based baseline that doesn't use cross-attention. Instead of cross-attention, this second baseline uses mean-pooling over the output of a Transformer encoder for encoding. For decoding, it uses the latent variable as a first token in a Transformer decoder, as is done for conditional generation with GPT-2 Santhanam & Shaikh (2019). These comparisons are performed using the same $\beta$-VAE objectives, and the decoder disentanglement scores as metrics. Training specifics and hyper-parameter settings are detailed in Appendix E. For each of the two baselines, the latent variables we vary during the decoder's evaluation are the mono-dimensional components of its latent vector. It is easier to pack information about the realizations of multiple syntactic roles into $D_z$ dimensions than into a single dimension. Consequently, the single dimensions we study for the baselines should be at an advantage to separate information into different variables.

Scoring disentanglement on the encoder side will not be possible for the baselines above as it requires attention values. To establish that our model effectively tracks syntactic roles, we compare it to a third baseline that locates each syntactic role through its median position across the dataset. This baseline is fairly strong on a language where word order is rigid (*i.e* configurational language) such as English. We refer to this Position Baseline as PB.

The scores are given for different values of $\beta$ (Eq. 1). Raising $\beta$ lowers the expressiveness of latent variables, but yields better disentanglement (Higgins et al., 2017). Following Xu et al. (2020), we set $\beta$ to low values to avoid posterior collapse. In our case, we observed that the models do not collapse for $\beta < 0.5$. Therefore, we display results for $\beta \in \{0.3, 0.4\}$. We stop at 0.3 as lower values for $\beta$ result in poorer generation quality. For our model we report performance for instances with $N_Z = 4$ (*ours-4*) and $N_Z = 8$ (*ours-8*).

**Results** The global disentanglement metrics are reported in Table 1.[9]

Table 1: Disentanglement quantitative results for the encoder (enc) and the decoder (dec). $N_\Gamma$ indicates the number of separated syntactic roles, and $\mathbb{D}$ measures concentration in a single variable. Values are averaged over 5 experiments. The standard deviation is between parentheses.

| Model | $\beta$ | $\mathbb{D}_{enc} \uparrow$ | $N_{\Gamma^{enc}} \uparrow$ | $\mathbb{D}_{dec} \uparrow$ | $N_{\Gamma^{dec}} \uparrow$ |
|---|---|---|---|---|---|
| Sequence VAE | 0.3 | - | - | 0.60(0.09) | 2.40(0.55) |
| | 0.4 | - | - | 1.28(0.24) | 1.40(0.55) |
| Transformer VAE | 0.3 | - | - | 0.12(0.10) | 3.00(0.70) |
| | 0.4 | - | - | 0.11(0.04) | 3.20(0.44) |
| PB | - | 0.98 (-) | 3.00(-) | - | - |
| ours-4 | 0.3 | 1.48(0.15) | 3.00(0.00) | 0.71(0.06) | 3.00(0.00) |
| | 0.4 | 1.43(0.79) | 3.00(0.00) | 0.72(0.37) | 2.80(0.45) |
| ours-8 | 0.3 | 1.34(0.18) | 3.80(0.45) | 0.51(0.14) | 2.80(0.45) |
| | 0.4 | 1.75(0.47) | 2.80(0.45) | 0.98(0.27) | 2.60(0.89) |

---

[8]`github.com/schmiflo/crf-generation/blob/master/generated-text/train`
[9]Fine-grained scores are given in Appendix F.

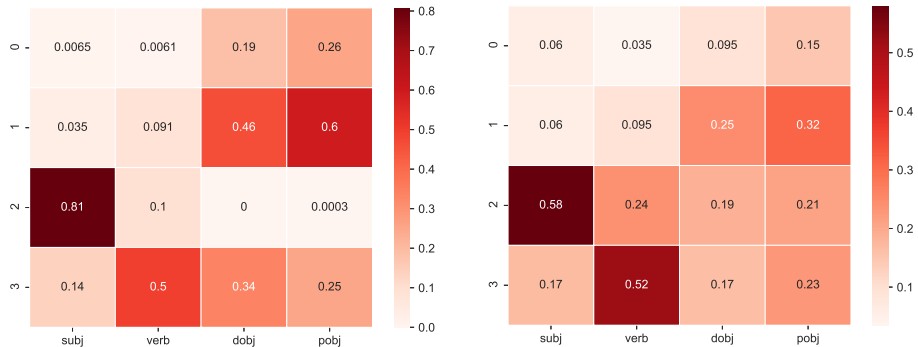

Figure 3: Encoder influence heatmap ($\Gamma^{\text{enc}}$). Figure 4: Decoder influence heatmap ($\Gamma^{\text{dec}}$).

Table 2: Resampling a specific latent variable for a sentence. The ID column is an identifier for the example.

| ID | Original sentence | Resampled subject | Resampled verb | Resampled dobj/pobj |
|---|---|---|---|---|
| 1 | people are sitting on the beach | a young man is sitting on the beach | people are playing in the beach | people are sitting on a bench |
| 2 | a man and woman are sitting on a couch | a man is sitting on a park bench | a man and woman are running on a grassy field | the man and woman are on a beach |
| 3 | a man is playing with his dog | a boy is playing in the snow | a man is selling vegetables | a man is playing the game with his goal . |

On the decoder side, the Sequence VAE exhibits disentanglement scores in the range of those reported for our model for $\beta = 0.3$, and higher for $\beta = 0.4$. However, $N_{\Gamma^{\text{dec}}}$ shows that it struggles to factor the realizations of different syntactic roles in different latent variables, and the higher score shown for $\beta = 0.4$ is accompanied by a lower tendency to separate the information from different syntactic roles. The Transformer VAE baseline assigns different latent variables to the different syntactic role (high $N_{\Gamma^{\text{dec}}}$), but suffers from very low specialization for these latent variables (low $\mathbb{D}_{dec}$). In contrast, our model is consistently able to separate 3 out of 4 syntactic roles, and while a higher $\beta$ raises its $\mathbb{D}_{dec}$, it does not decrease its $N_{\Gamma^{\text{dec}}}$. As *ours-8* has more latent variables, this encourages the model to further split the information in each syntactic role between more latent variables[10]. The fact that ADVAEs perform better than both Sequence VAEs and classical Transformer VAEs shows that its disentanglement capabilities are due to the usage of Cross-Attention to obtain latent variables, and not only to the usage of Transformers. On the encoding side, our models consistently score above the baseline, showing that our latent variables actively follow the syntactic roles.

In Figures 3 and 4, we display the influence matrices $\Gamma^{\text{enc}}$ and $\Gamma^{\text{dec}}$ for an instance of our ADVAE with $N_Z = 4$ as heatmaps. The vertical axes correspond to the latent variables. As can be seen, our model successfully associates latent variables to verbs and subjects but chooses not to separate direct objects and prepositional objects into different latent variables. Upon further inspection of the same heatmaps for the VAE baseline, it appears that it most often uses a single latent variable for *verb* and *subj*, and another for *dobj* and *pobj*.

One can also notice in Figures 3 and 4, that the encoder matrix is sparser than the decoder matrix (which is consistent with the higher encoder disentanglement scores in Table 1). This is to be expected as the decoder $p_\theta(s|z)$ adapts the realizations of syntactic roles to each other after they are sampled separately from $p(z)$. The reason for this is that the language modeling objective requires some coherence between syntactic roles (conjugating verbs with subjects, changing objects that are semantically inadequate for a verb, etc). This *co-adaptation*, contradicts the independence of our latent variables. It will be further discussed in the following paragraph.

---

[10]Results for a larger grid of $N_z$ values are reported in Appendix K, and show that latent variables still clearly relate to syntactic roles, but in groups.

Table 3: Swapping the value of a specific latent variable between two sentences. The SSR (Swapped Syntactic Role) column indicates the syntactic role that has been swapped.

| ID | Sentence 1 | Sentence 2 | SSR | Swapped Sentence 1 | Swapped Sentence 2 |
|----|-----------|-----------|-----|-------------------|-------------------|
| 1 | a woman is talking on a train | people are sitting on the beach | subj | people are talking on a train | a woman is sitting on the beach |
| 2 | people are sitting on the beach | a woman is talking on a train | verb | people are talking on a beach | a woman is standing on a train |
| 3 | a woman is talking on a train | a man is playing with his dog | dobj/ pobj | a man is playing the guitar with a goal | a woman is performing a trick |

**Changing the Realizations of Syntactic Roles**    Here, we display of few qualitative examples of how the realizations of syntactic roles can be separately changed using an instance of our ADVAE.

As a first example, we generate a sentence from a random latent vector, then resample for each syntactic role the corresponding disentangled latent variable to observe the change on the subsequently generated altered sentence. The results of this manipulation are in Table 2[11]. As can be seen, some examples exhibit changes that only affect the target syntactic role (example 1). However, the model often produces co-adaptations that go past the target syntactic role either for semantic soundness (example 2, resampled verb adapts the object), or simply for lack of generalization from the SNLI data used for training.

A second example we display is a swap of syntactic role realizations between sentences. A few examples are given in Table 3. Similar to Table 2, the model often yields the expected result. Co-adaptation is best seen here, as taking a syntactic role to a sentence with which it is incompatible results in unexpected changes (example 3).

**Further investigations**    As this is a first step in this research direction, we conducted this study on a dataset of relatively regular sentences. Running similar experiments on a dataset with more complicated and diverse sentence structures such as in Yelp (Appendix B) results in the same comparative patterns. However, disentanglement scores are much lower. This calls for future iterations to improve upon ADVAE and our evaluation protocol to better model structure in order to scale to User Generated Content (UGC). Our experiments also enabled underlining an inherent issue to syntactic role disentanglement: *co-adaptation*. The independence between our latent variables causes the decoder $p_\theta(s|z)$ to correct the incoherence between independently sampled syntactic role realizations. Using structured latent variables to learn relations between syntactic roles seems to be the natural solution to this problem. An investigation of a hierarchical version of the ADVAE (Appendix A) showed, however, that a drop-in replacement of the independent prior with a structured prior is not sufficient in order to *absorb* co-adaptation into the latent variable model. Our future works will, therefore, also include the investigation of training techniques that can achieve improved results with structured latent variables.

## 7    CONCLUSION

We introduce a framework to study the disentanglement of syntactic roles and show that it is possible to learn a representation of sentences that exhibits separation in the realizations of these syntactic functions *without supervision*. Our framework includes: *i)* Our model, the ADVAE, which maps syntactic roles to separate latent variables more often than standard Sequence VAEs and with better concentration than standard Transformer VAEs, and allows for the use of attention to study the interaction between latent variables and spans, *ii)* An evaluation protocol to quantify disentanglement between latent variables and spans both in the encoder and in the decoder.

Our study constitutes a first step in a promising process towards *unsupervised* explainable modeling and fine-grained control over the predicate-argument structure of sentences. Although we focused on syntactic roles realizations, this architecture as well as the evaluation method are generic and could be applied to other tasks. The architecture could be used at the document level (*e.g.* disentangling discourse relations), while the evaluation protocol could be applied to other spans such as constituents.

---

[11]More Examples are available in Appendix H

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

## A  A HIERARCHICAL VERSION OF OUR ADVAE

As we stated, our ADVAE aims to factor sentences into independent latent variables. However, given the dependency structure of sentences, realizations of syntactic roles are known to be interdependent to some degree in general. Therefore one may think that a structured latent variable model would be better suited to model the realizations of syntactic roles. In fact, such a model could absorb the language modeling co-adaptation between syntactic roles. For instance, instead of sampling an object and a verb from $p(z)$ that are inadequate, then co-adapting them through $p_\theta(s|z)$, a structured $p_\theta(z)$ could produce an *adequate* object for the verb. For this experiment, rather than using an independent prior $p(z)$, we use a structured prior $p_\theta(z) = p(z^0) \prod_{l=1}^{L} p_\theta(z^l|z^{l-1})$ where $p(z^0)$ is a standard Gaussian, and all subsequent $L-1$ hierarchy levels are parameterized by learned conditional diagonal Gaussians. The model used for each $p_\theta(z^l|z^{l-1})$ is shown in Figure 5 below:

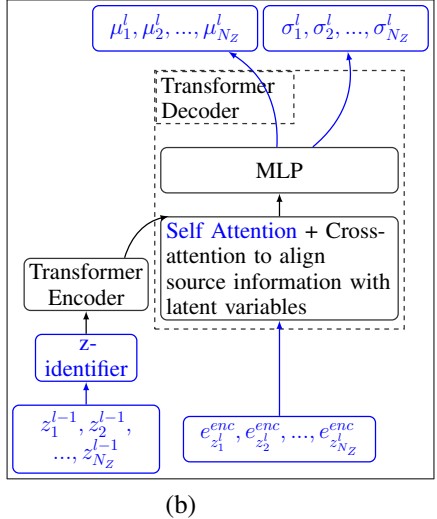

(b)

Figure 5: The conditional inference module linking each of the hierarchy levels in our prior with the next level $p_\theta(z^l|z^{l-1})$. This module treats latent variables from previous layers as they are treated in our original decoder, and generates parameters for latent variables in subsequent hierarchy levels as it is done in our encoder.

We display the results for $L = 2$ and $L = 3$ in Table 4. For both models, we set $N_Z$ to 4.

Table 4: Disentanglement results for structured latent variable models on SNLI.

| Depth | $\beta$ | $\mathbb{D}_{enc}$ | $N_{\Gamma^{enc}}$ | $\mathbb{D}_{dec}$ | $N_{\Gamma^{dec}}$ |
|---|---|---|---|---|---|
| $L=2$ | 0.3 | 0.79(0.36) | 3.60(0.55) | 0.51(0.22) | 2.60(0.55) |
| | 0.4 | 0.42(0.23) | 2.80(0.45) | 0.12(0.20) | 2.20(0.45) |
| $L=3$ | 0.3 | 0.90(0.25) | 3.14(0.69) | 0.52(0.20) | 2.43(0.53) |
| | 0.4 | 0.32(0.38) | 2.75(0.50) | 0.25(0.42) | 2.25(0.50) |

The results show lower mean disentanglement scores, and high standard deviations compared to the standard version of our ADVAE. By inspecting individual training instances of this hierarchical model, we found that some instances achieve disentanglement with close scores to those of the standard ADVAE, while others completely fail (which results in the high variances observed in Table 4). Unfortunately, hierarchical latent variable models are notoriously difficult to train (Zhao et al., 2017). Our independent latent variable model is therefore preferable to the structured one due to these empirical results. More advanced hierarchical latent variable training techniques (such as Progressive Learning and Disentanglement (Li et al., 2020b)) may, however, provide better results.

## B  EXPERIMENTING WITH THE YELP DATASET

We investigated the behavior of our ADVAE of on the user-generated reviews from the Yelp dataset used in Li et al. (2018) using the same procedure we used for SNLI. The length of sentences from this dataset ($8.88 \pm 3.64$) is similar to the length of sentences from the SNLI dataset. Similar to the experiments in the main body of the paper, we display the disentanglement scores in Table 5, and the influence metrics of one of the instances of our model as heatmaps in Figures 6 and 7.

Table 5: Disentanglement results for the Yelp dataset

| Model | $\beta$ | $\mathbb{D}_{enc}$ | $N_{\Gamma^{enc}}$ | $\mathbb{D}_{dec}$ | $N_{\Gamma^{dec}}$ |
|---|---|---|---|---|---|
| Sequence VAE | 0.3 | - | - | 0.44(0.09) | 2.20(0.45) |
| | 0.4 | - | - | 1.21(0.06) | 2.25(0.50) |
| PB | - | 0.33(-) | 2.00(-) | - | - |
| ours-4 | 0.3 | 0.48(0.07) | 2.00(0.00) | 0.18(0.02) | 2.50(0.58) |
| | 0.4 | 0.54(0.04) | 3.00(0.00) | 0.23(0.03) | 2.40(0.55) |
| ours-8 | 0.3 | 0.44(0.04) | 3.80(0.45) | 0.17(0.04) | 2.80(0.84) |
| | 0.4 | 0.57(0.26) | 3.40(0.55) | 0.15(0.10) | 2.40(0.89) |

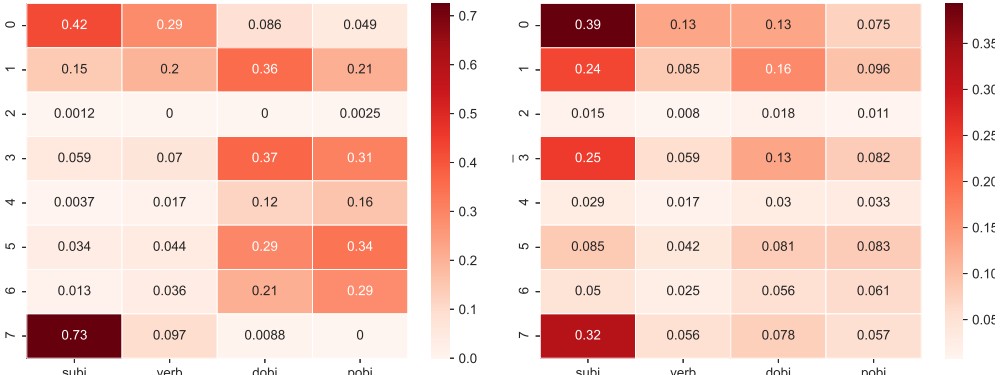

Figure 6: Encoder influence heatmap for Yelp($\Gamma^{enc}$).

Figure 7: Decoder influence heatmap for Yelp($\Gamma^{dec}$).

Although the results show similar trends, they are weaker than what we obtained for SNLI. Given the difference between SNLI and Yelp (displayed in Appendix D) there are two clear reasons for this decrease. The first is that Yelp is a dataset where it is harder to locate the syntactic roles. This is illustrated by the fact that the PB baseline obtains a much lower score. The second is that our syntactic role extraction heuristics are tailored for regular sentences with verbal roots, which subjects the evaluation metrics on Yelp to a considerable amount of noise. Nevertheless, the comparisons between a VAE, an ADVAE, and PB retain the same conclusions, but with lower margins and some overlapping standard deviations.

Through manual inspection of examples, we observed that the various structural characteristics (enumerations, sentences with nominal roots, presence of coordinating conjunctions, etc) were captured by different variables. This indicates that future iterations of our model need to provide ways to separate structural information from content-related information.

## C  MEASURING THE EFFECT OF LATENT VARIABLES ON THE STRUCTURE OF SENTENCES

In Figure 8, for each latent variable and each syntactic role, we report the probability that resampling the latent variable causes the appearance/disappearance of the syntactic role. The instance we use here is the same as the one we use for the heatmaps in the main body of the paper. According to the heatmaps in Figures 3 and 4, latent variable 3 is the one associated with the verb. As can be seen in the present heatmap in Figure 8, this same variable is the one that has the most influence on the appearance/disappearance of direct and prepositional objects, and this is a pattern that proved to be consistent across our different runs. This constitutes empirical justification for our choice of discarding these cases from our decoder influence metrics.

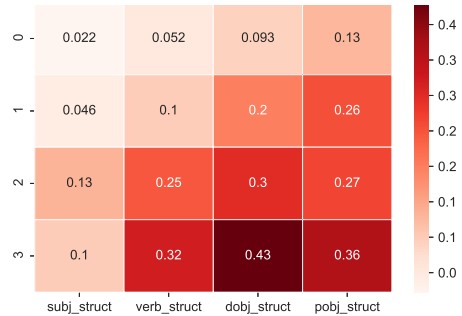

Figure 8: The influence of latent variables on the appearance or disappearance of syntactic roles.

## D    EXAMPLE SENTENCES FROM YELP AND SNLI AND THEIR CORRESPONDING SYNTACTIC EXTRACTIONS

Table 6 shows some samples from SNLI and Yelp reviews. Samples from Yelp Reviews exhibit a clearly higher structural diversity. On the other hand, most SNLI samples are highly similar in structure.

Our syntactic role extraction heuristics were tailored for sentences with verbal roots. As a result, it can be seen that they struggle with sentences with nominal roots as well as other forms of irregular utterances present in Yelp. For SNLI, our extractions mostly yield the expected results, allowing for a reliable global assessment of our models.

Table 6: Example syntactic role extractions from both SNLI and Yelp

| Source | Sentence | subj | verb | dobj | pobj |
|--------|----------|------|------|------|------|
| Yelp | i was originally told it would take _num_ mins . | it | told | _ num _ mins | |
| Yelp | slow , over priced , i 'll go elsewhere next time . | i | go | | |
| Yelp | we will not be back | we | | | |
| Yelp | terrible . | | | | |
| Yelp | at this point they were open and would be for another hour . | they | | | this point |
| SNLI | people are outside playing baseball . | people | | baseball | |
| SNLI | two dogs pull on opposite ends of a rope . | two dogs | pull | opposite ends of a rope | a rope |
| SNLI | a lady lays at a beach . | a lady | lays | | a beach |
| SNLI | people are running through the streets while people watch . | people | running | | the streets |
| SNLI | someone prepares food into bowls | someone | prepares | food | bowls |

## E    TRAINING DETAILS AND HYPER-PARAMETER SETTINGS

**Our ADVAE's hyper-parameters**   Our model has been set to be large enough to reach a low reconstruction error during the initial reconstruction phase of the training. We use 2-layer Transformers with 4 attention heads and a hidden size of 192. Contrary to Vanilla VAEs, our model seems to perform better with high values of $N_Z$. Therefore, we set our latent vector to a size of 768, and divide it into 96-dimensional variables for our $N_Z = 8$ model and to 192-dimensional latent variables for our $N_Z = 4$ model. No automated hyper-parameter selection has been done afterward.

**Sequence VAE hyper-parameters**   As is usually done for this baseline (Xu et al., 2020), we set both the encoder and the decoder to be 2-layer LSTMs.
We run this model for hidden LSTM sizes in [256, 512], and latent vector sizes in [16, 32]. The results for the model scoring the highest $\mathbb{D}_{dec}$ are then reported. Even though selection has been done according to $\mathbb{D}_{dec}$, we checked the remaining instances of our baselines and they also yielded low $N_{\Gamma^{dec}}$ values.

**Transformer VAE hyper-parameters**   We set the hidden sizes and number of layers for this baseline similarly to ADVAE, since it is also a Transformer. We run this model for latent vector sizes in [16, 32] and display the highest scoring model, as is done for the Sequence VAE.

**Training phases**   All our models are trained using ADAM(Kingma & Ba, 2015) with a batch size of 128 and a learning rate of 2e-4 for 20 epochs. The dropout is set to 0.3. To avoid posterior collapse, we train all our models for 3000 steps with $\beta = 0$ (reconstruction phase), then we linearly increase $\beta$ to its final value for the subsequent 3000 steps. Following Bowman et al. (2016), we also use word-dropout. We set its probability to 0.1.

**Evaluation**   For the evaluation, $T^{dec}$ is set to 2000, and $T^{enc}$ is equal to the size of the test set.

## F   DISENTANGLEMENT SCORES FOR EACH SYNTACTIC ROLE

The full disentanglement scores are reported in Table 7 for the decoder, and in Table 8 for the encoder.

Table 7: Complete decoder disentanglement scores for SNLI

| Model | $\beta$ | $\mathbb{D}_{dec}$ | $N_{\Gamma^{dec}}$ | $\Delta\Gamma_{dec,verb}$ | $\Delta\Gamma_{dec,subj}$ | $\Delta\Gamma_{dec,dobj}$ | $\Delta\Gamma_{dec,pobj}$ |
|---|---|---|---|---|---|---|---|
| | 0.3 | 0.68(0.22) | 2.80(0.45) | 0.19(0.04) | 0.35(0.18) | 0.06(0.03) | 0.07(0.03) |
| ours-4 | 0.4 | 0.81(0.05) | 3.00(0.00) | 0.21(0.04) | 0.47(0.03) | 0.06(0.02) | 0.07(0.02) |
| | 0.3 | 0.60(0.10) | 3.00(0.00) | 0.17(0.04) | 0.31(0.08) | 0.05(0.04) | 0.07(0.04) |
| ours-8 | 0.4 | 0.63(0.35) | 2.80(0.45) | 0.17(0.10) | 0.32(0.18) | 0.05(0.04) | 0.08(0.05) |
| | 0.3 | 0.60(0.09) | 2.40(0.55) | 0.24(0.06) | 0.03(0.04) | 0.03(0.02) | 0.31(0.03) |
| Sequence VAE | 0.4 | 1.28(0.24) | 1.40(0.55) | 0.45(0.12) | 0.23(0.02) | 0.02(0.02) | 0.57(0.11) |
| | 0.3 | 0.12(0.10) | 3.00(0.70) | 0.01(0.01) | 0.07(0.06) | 0.01(0.01) | 0.03(0.03) |
| Transformer VAE | 0.4 | 0.11(0.04) | 3.20(0.44) | 0.03(0.02) | 0.04(0.04) | 0.01(0.01) | 0.02(0.01) |

Table 8: Complete encoder disentanglement scores for SNLI

| Model | $\beta$ | $\mathbb{D}_{enc}$ | $N_{\Gamma^{enc}}$ | $\Delta\Gamma_{enc,verb}$ | $\Delta\Gamma_{enc,subj}$ | $\Delta\Gamma_{enc,dobj}$ | $\Delta\Gamma_{enc,pobj}$ |
|---|---|---|---|---|---|---|---|
| | 0.3 | 1.30(0.09) | 3.00(0.00) | 0.28(0.05) | 0.65(0.02) | 0.08(0.03) | 0.29(0.03) |
| ours-4 | 0.4 | 1.46(0.33) | 3.00(0.00) | 0.38(0.12) | 0.64(0.10) | 0.14(0.04) | 0.30(0.10) |
| | 0.3 | 1.36(0.13) | 3.40(0.89) | 0.44(0.12) | 0.60(0.18) | 0.21(0.08) | 0.11(0.06) |
| ours-8 | 0.4 | 1.44(0.79) | 3.40(0.55) | 0.42(0.23) | 0.61(0.34) | 0.17(0.10) | 0.23(0.16) |
| Average Position | - | 0.98 (-) | 3.00(-) | 0.12(-) | 0.70(-) | 0.12(-) | 0.04(-) |

## G   DISENTANGLEMENT HEATMAPS OVER THE ENTIRE RANGE OF SYNTACTIC ROLES AND POS TAGS

We report decoder and encoder heatmaps for all the syntactic roles following the Stanford Dependencies (SD; De Marneffe & Manning, 2008) annotation scheme of Ontonotes, which was used to train our Spacy2 parser, in Figures 9 and 10. For the sake of extensiveness and to make sure we did not draw results from some parser biases, we also report the same heatmaps but using UDPipe 2.0 (Straka, 2018), which uses UD type annotations[12], in Figures 13 and 14. Finally, we also report

---

[12]A widely adopted annotation scheme derived from Stanford Dependencies.

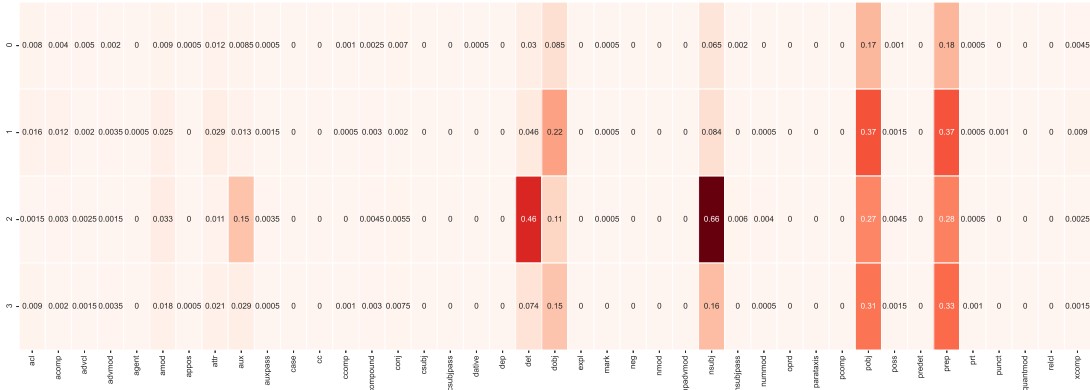

Figure 9: Decoder influence heatmap for all SD syntactic roles.

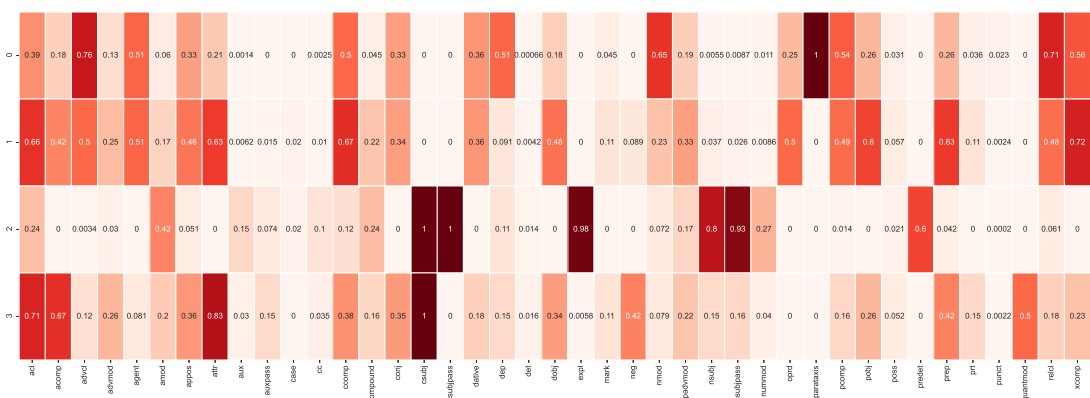

Figure 10: Encoder influence heatmap for all SD syntactic roles.

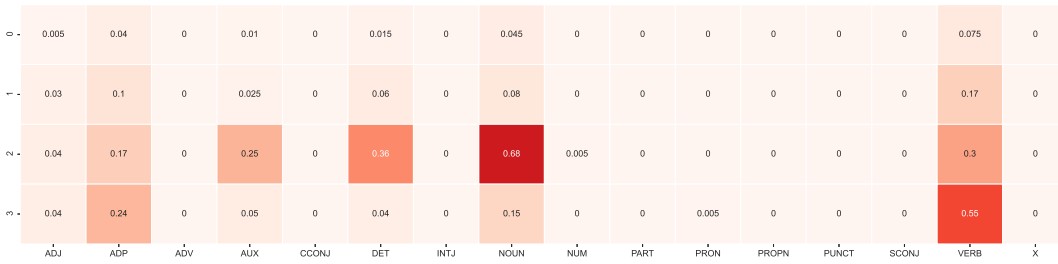

Figure 11: Decoder influence heatmap for all PoS Tags.

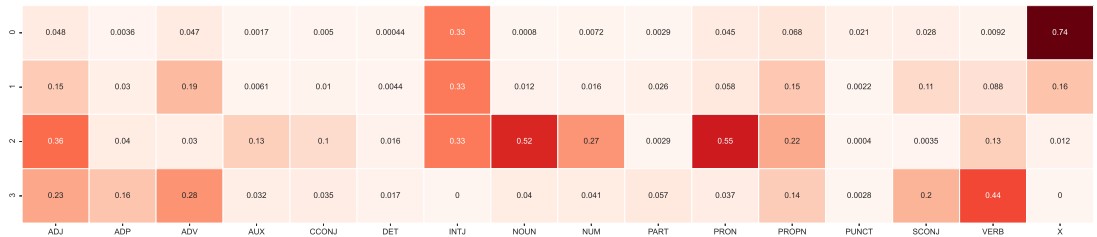

Figure 12: Encoder influence heatmap for all PoS Tags.

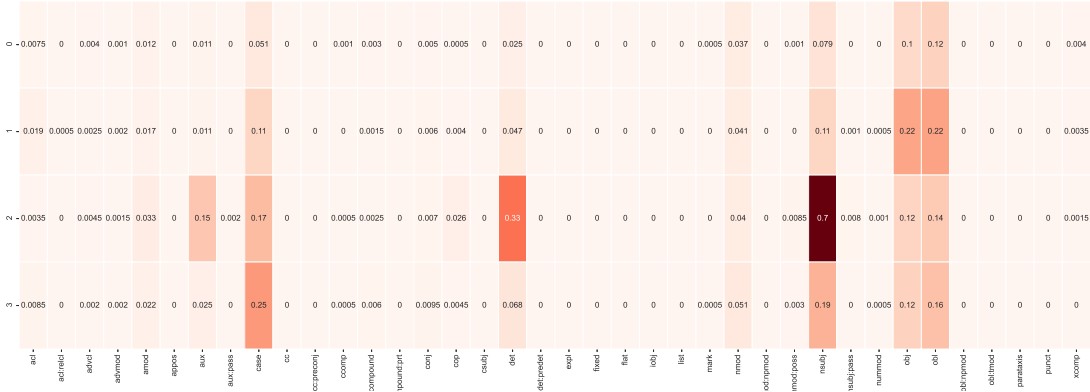

Figure 13: Decoder influence heatmap for all UD syntactic Roles.

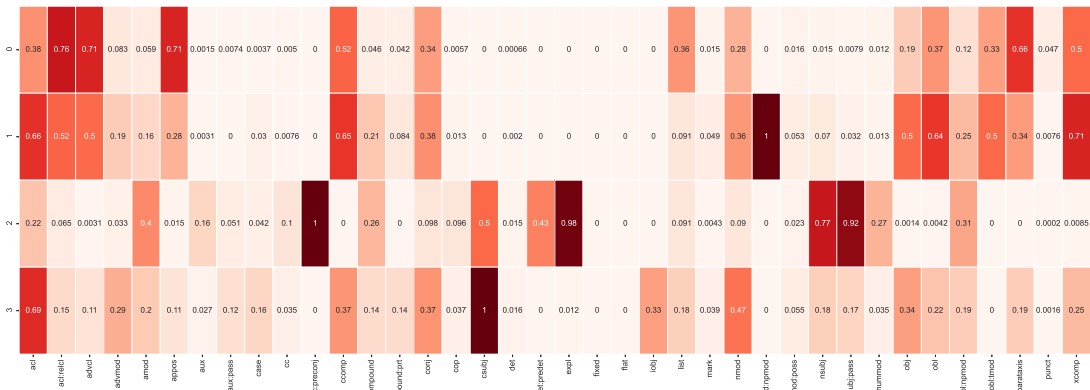

Figure 14: Encoder influence heatmap for all UD syntactic Roles.

heatmaps for interaction with PoS Tags extracted with Spacy2 in Figures 11 and 12. As was done in the main body of the paper, the span corresponding to each syntactic role (in both annotation schemes) was taken to be the series of words included in its corresponding subtree. In contrast, the span corresponding to each PoS tag was just taken to be the tagged word. Results from UD parsing extraction lead to the same conclusions as from our initial SD results.

The instance of our ADVAE for which we display the above heatmaps is the same one for which we display the heatmaps in Figures 3 and 4 in the main body of the paper. As shown in those Figures, it mostly uses variable 3 for verbs, variables 2 for subjects, and variable 1 for objects. The remaining variable (0) also seems to capture some interaction with objects. The heatmaps show that our ADVAE tends to group syntactic roles into latent variables in a way that aligns with the predicative structure of sentences. In fact, variable 2 displays the highest influence on the PoS tag VERB as well as its surroundings as a predicate argument such as adverbs and adverbial phrases. Similarly, latent variable 2 displays a high influence on subjects (nominal or clausal), numeral modifiers, adjectival modifiers, and auxiliaries (for conjugation). Moreover, Variable 1 highly influences the direct and prepositional objects, which we study in the main body of the paper, but also diverse clausal modifiers and obliques which often play similar roles to direct and prepositional objects in a predicate structure.

# H  ADDITIONAL EXAMPLES OF RESAMPLED REALIZATIONS FOR EACH SYNTACTIC ROLE

Table 9 contains a wide array of examples where the latent variable corresponding to each syntactic role is resampled.

Table 9: More examples where we resample a specific latent variable for a sentence.

| Original sentence | Resampled subject | Resampled verb | Resampled dobj/pobj |
|---|---|---|---|
| the woman is riding a large brown dog | two men are riding in a large city | the woman is wet | the woman is riding on the bus |
| the police are running in a strategy | a man is looking at a date | the police are at an arid | the police are running in a wooded area |
| a man is holding a ball | a man is holding a ball | a man is , and a woman are talking on a road | a man is sitting on a cell-phone outside |
| everyone is watching the game | some individuals are watching tv | everyone is a man | everyone is watching the game in the air |
| there is a man in the air | a man is sitting in the air | there is no women wearing swim trunks | there is a man in a red shirt |
| a group of friends are standing on a beach | an elderly father and child are standing on the beach | a group of people are standing on a beach | a group of friends are looking at the beach |
| the women are in a store | a man is playing a game | two women are on a break | two women are sitting on a bench |
| a man is playing a game | a little girl is playing with a ball | a man is clean | a man is sitting on a lake to an old country |
| a man is playing a game | some dogs are playing in the pool | a man is preparing to chase himself | a man is playing a game |
| the memorial woman is happy | a dog is happy | the memorial workers are in a room | the memorial is happy |
| a man is wearing a green jacket and a ship | a boy sitting in a green device | a man is dancing for the camera | the man is wearing a hat |
| a man is playing a game | a man is playing a game | two men are tripod | a man is playing with a guitar |
| a man is wearing a brown sweater and green shirt | a karate dog is swimming in a chair | a man is bought a brown cat in an airplane | a man is wearing a dress and talks to the woman |
| the woman is about to visitors | three people are working at a babies | the woman is wearing a sewer | the woman is about to sell a tree |
| a man is sitting in the snowy field | a man is sitting in the snowy field | a man is wearing electronics | a man is sitting on a park bench |
| two people are playing in the snow | the motorcycle is a woman on the floor | two people play soccer in the snow | two people are playing in a concert |
| a man is standing next to another man | a boy is standing next to another man | a man is standing | a man is standing next to a man |
| a man is on his bike | a man is on his bike | a dog is showing water | a man is on his bike |
| a man is sitting in front of a tree , taking a picture | a man is sitting in front of a tree | a man is holding a red shirt and climbing a tree | a man is sitting on a suburban own |
| a man is sitting with a dog | the children are sitting with the dog | a man is playing with a dog | a man is sitting with an umbrella |
| the man is holding a ball | a boy is playing with a ball | the man is on a bike | the man is waiting for a counts to jump for the first base |
| a man is holding a game | five people buying a skateboard from easter | a man is on a bicycle | a man is very large |
| two men are playing in a field | the kids play in the snow | two men are playing a game | two men are playing in a field |
| a man is wearing a hat | a woman is wearing a hat | the man is they oil | a man is wearing a black bathing suit near buildings |
| a man is playing a guitar | the man is wearing a blue shirt | a man is sitting on a bench | a man wearing a hat is playing a guitar |
| a woman is playing a game | a man is playing a game | a woman is playing a game | a woman is playing a game |
| a man is on the truck | the people are on the truck | a man is holding a truck | a man is on the grass |
| a man is playing with the cut | a small boy is playing on the cut | a man is warming up the cut | a man is playing a game |
| a group of people are at a park | the man is wearing a blue shirt | a group of people are at a park | a group of people are at a park |

## I RECONSTRUCTION AND KULLBACK-LEIBLER VALUES ACROSS EXPERIMENTS

Table 10: Reconstruction loss and Kullback-Leibler values.

| Model | $\beta$ | $-\mathbb{E}_{(z)\sim q_\phi(z\|x)}\left[\log p_\theta(x\|z)\right]$ | $D_{\mathrm{KL}}[q_\phi(z\|x)\|p(z)]$ | Perplexity Upper Bound |
|---|---|---|---|---|
| Sequence VAE | 0.3 | 31.38(0.12) | 2.80(0.25) | 22.02(0.30) |
| | 0.4 | 32.19(0.13) | 1.22(0.04) | 21.08(0.22) |
| Transformer VAE | 0.3 | 24.35(0.14) | 13.38(0.19) | 25.07(0.27) |
| | 0.4 | 26.57(0.27) | 8.36(0.32) | 20.68(0.16) |
| ours-4 | 0.3 | 10.75(0.94) | 42.63(1.16) | 68.49(5.96) |
| | 0.4 | 16.01(0.64) | 27.93(1.52) | 36.16(2.20) |
| ours-8 | 0.3 | 8.83(1.66) | 46.99(2.99) | 77.26(9.02) |
| | 0.4 | 16.84(8.50) | 27.34(14.99) | 39.23(11.27) |

The values for the reconstruction loss, the $D_{\mathrm{KL}}$ divergence, and the upper bound on perplexity concerning the experiments in the main body of the paper are reported in Table 10. Since our models are VAE-based, one can only obtain the upper bound on the perplexity and not its exact value. These upper bound values are obtained using an importance sampling-based estimate of the negative log-likelihood, as was done in Wu et al. (2020). We set the number of importance samples to 10.

It can be seen that the behavior of ADVAEs is very different from classical Sequence VAEs and Transformer VAEs. On the plus side, they are capable of sustaining much more information in their latent variables as shown by their higher $D_{\mathrm{KL}}$, and they do better at reconstruction. The upper bound estimate of their perplexity is however higher. A high $D_{\mathrm{KL}}$ makes it more difficult for the importance sampling-based perplexity estimate to reach the true value of the model's perplexity. This may be the reason behind the higher values observed for ADVAEs.

## J LAYER-WISE ENCODER ATTENTION

In the main body of the paper, we use attention values that are averaged throughout the network. We hereby display the encoder heatmaps obtained by using attention values from the first layer (Fig. 15), the second layer (Fig. 16), or an average on both layers (Fig. 17) for comparison.

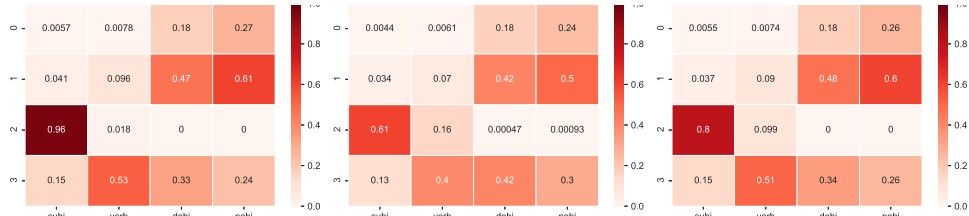

Figure 15: Encoder influence heatmap ($\Gamma^{\mathrm{enc}}$) when only using the *first* layer.

Figure 16: Encoder influence heatmap ($\Gamma^{\mathrm{enc}}$) when only using the *second* layer.

Figure 17: Encoder influence heatmap ($\Gamma^{\mathrm{enc}}$) when *averaging* over both layers.

As can be seen, the first layer alone provides the most sparse heatmap, and thus, the clearest correspondence between syntactic roles and latent variables. This corroborates the claims of Tenney et al. (2020) about syntax being most prominently processed in the early layers of Transformers.

## K ADVAE RESULTS FOR A LARGER GRID OF $N_z$ VALUES

We display in Table 11 the quantitative results of ADVAE on SNLI for $N_z$ in $\{2, 4, 6, 8\}$. For ours-2, it is normal that it only separates syntactic role realizations into a maximum of 2 latent variables, as seen from the values of $N_{\Gamma^{\mathrm{enc}}}$ and $N_{\Gamma^{\mathrm{enc}}}$, since 2 is its total number of latent variables.

As observed in the main body of the paper, the increase of the number of latent variables used in ADVAE leads to dispatching the influence on the realization of a single syntactic role to multiple latent variables. This in turn,leads to for $\mathbb{D}_{enc}$ and $\mathbb{D}_{dec}$ that we observe . In Figures 18 and 19, we

respectively display the encoder and decoder heatmaps of ADVAE with 16 latent variables. As can be seen in these figures, each latent variable still highly specializes in a specific syntactic role. This is seen more clearly on the encoder heatmap due to co-adaptation harming the clarity of the decoder heatmap. This specialization seems to be shared among groups (*e.g.* variables 4 and 8 specialize in the subject, as indicated by the green squares on the figure). This causes the difference of influence between the most influential variable and the second most influential one to be low, and thus decreases the values of $\mathbb{D}_{enc}$ and $\mathbb{D}_{dec}$.

Table 11: Disentanglement quantitative results on SNLI for a larger grid of $N_z$ values.

| Model | $\beta$ | $\mathbb{D}_{enc}$ | $N_{\Gamma^{enc}}$ | $\mathbb{D}_{dec}$ | $N_{\Gamma^{dec}}$ |
|---|---|---|---|---|---|
| ours-2 | 0.3 | 2.01(0.07) | 2.00(0.00) | 0.92(0.06) | 2.00(0.00) |
| | 0.4 | 0.33(0.15) | 1.60(0.55) | 0.13(0.09) | 1.20(0.45) |
| ours-4 | 0.3 | 1.30(0.09) | 3.00(0.00) | 0.68(0.22) | 2.80(0.45) |
| | 0.4 | 1.46(0.33) | 3.00(0.00) | 0.81(0.05) | 3.00(0.00) |
| ours-8 | 0.3 | 1.36(0.13) | 3.40(0.89) | 0.60(0.10) | 3.00(0.00) |
| | 0.4 | 1.44(0.79) | 3.40(0.55) | 0.63(0.35) | 2.80(0.45) |
| ours-16 | 0.3 | 0.60(0.31) | 3.60(0.55) | 0.33(0.30) | 2.60(0.55) |
| | 0.4 | 0.65(0.16) | 3.40(0.55) | 0.56(0.28) | 2.60(0.55) |

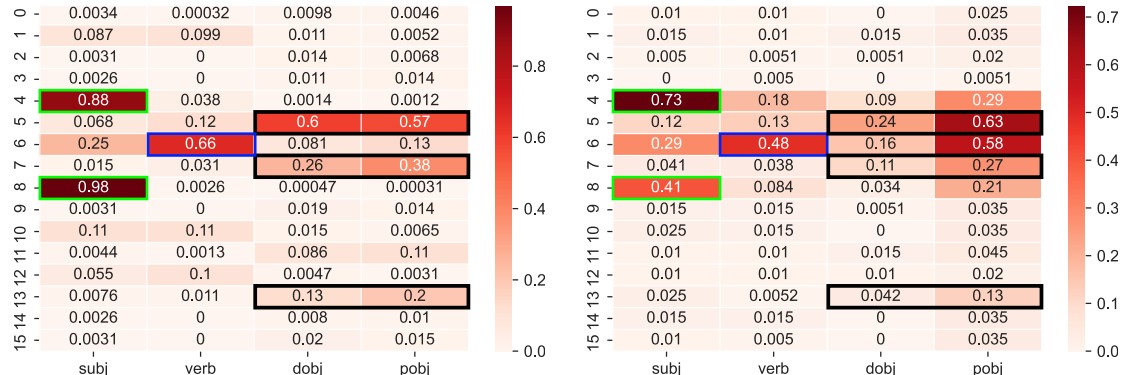

Figure 18: Encoder influence heatmap for ADVAE with 16 latent variables on SNLI ($\Gamma^{enc}$). Squares with similar colors highlight groups of latent variables that relate to the same syntactic role.

Figure 19: Decoder influence heatmap for ADVAE with 16 latent variables on SNLI ($\Gamma^{dec}$). Squares with similar colors highlight groups of latent variables that relate to the same syntactic role.

