# OpenReview forum: "Towards Unsupervised Content Disentanglement in Sentence Representations via Syntactic Roles"
_ICLR.cc/2022/Conference — ICLR 2022 Submitted_

### Official Review · Reviewer_WuPD · 2021-11-01

**Correctness:** 2
**Technical Novelty And Significance:** 3
**Empirical Novelty And Significance:** 2
**Recommendation:** 5
**Confidence:** 3

**Main Review:**

Pros:
- The insight that cross-attention disentangles content automatically in Transformers is interesting, and I feel that this insight could be exploited for a variety of controllable text generation tasks.

Cons:
- The baseline VAE is not a proper comparison, given that an LSTM is used (versus a Transformer in the proposed model). Please either make the proposed model LSTM-based or the baseline a Transformer.
- I feel that the ADVAE, while explained relatively clearly in the figure, could benefit from a step-by-step explanation in the text. I found it slightly confusing to follow section 3, which could benefit from a re-organization: for example, 1st introduce the operations happening in Fig 2b, describe the latent variable generation procedure, and then what happens in Fig 2c (reconstruction). An example of a paper that I feel does this well is Zhang et al., 2016.
- I also feel that the impact of increasing/decreasing N_z is not discussed enough (e.g. if you increase the N_z to become the maximum sequence length, what happens, etc…).
- Overall, I feel it would be pertinent to test how well different aspects (not just syntactic roles) are disentangled. Rather than the method per se, I feel that this insight into unsupervised disentanglement is interesting, however, constraining the method to syntactic roles can be limiting. For example, for the “low complexity” datasets (e.g. sentiment analysis), it would be interesting to see when running your method, is there a vector that emerges that can control sentiment, or is it able to both disentangle syntactic roles and sentiment, etc….? (As far as I know, John et al. 2019, did this in a supervised manner).

Minor comments/missing references:
- Machine Translation/MT Transformer -> Transformer encoder-decoder models (as this architecture is widely used, even in non-MT tasks)
- Sentence representations have been extracted from sequence-to-sequence Transformer models before (Lewis et al., 2020, Raffel et al., 2020, Siddhant et al., 2020)
- The [CLS] representation is generally the default sentence representation in BERT (not [SEP])
- p_r (the dependency parser?) is not defined.

Questions/Other comments:
- It would be interesting to compare the variational model to a denoising AE (seq2seq masked language model; Lewis et al., 2020), in which the input is corrupted and the prior distribution is removed). This could make the method more generally applicable, even to large pre-trained models.
- Furthermore, I am sceptical of the use of self-attention when computing the mean and standard deviation vectors (Fig 1b). I would be curious what would happen if only cross-attention would be used. If that is the case, then the argument could be made that the original e^{enc} vectors correspond to a syntactic role (or other factors) themselves.
- Also, it would be curious if this emerges in machine translation (given the motivation stated as such), where the source and target represent the same sentence in different languages.
---

Lewis, Mike et al. “BART: Denoising Sequence-to-Sequence Pre-Training for Natural Language Generation, Translation, and Comprehension.” Proceedings of the 58th Annual Meeting of the Association for Computational Linguistics (2020): n. pag. Crossref. Web.

Siddhant, Aditya et al. “Evaluating the Cross-Lingual Effectiveness of Massively Multilingual Neural Machine Translation.” Proceedings of the AAAI Conference on Artificial Intelligence 34.05 (2020): 8854–8861. Crossref. Web.

John, Vineet et al. “Disentangled Representation Learning for Non-Parallel Text Style Transfer.” Proceedings of the 57th Annual Meeting of the Association for Computational Linguistics (2019): n. pag. Crossref. Web.

Raffel, Colin et al. “Exploring the Limits of Transfer Learning with a Unified Text-to-Text Transformer.” ArXiv abs/1910.10683 (2020): n. pag.

Zhang, Biao et al. “Variational Neural Machine Translation.” Proceedings of the 2016 Conference on Empirical Methods in Natural Language Processing (2016)

**Summary Of The Paper:**

This paper proposes a new model ADVAE, which uses a sequence of latent variables which are constructed using cross-attention, which are then used to condition the inference model. The findings show that certain latent variables correlated with different syntactic roles (measured using a dependency parser).  The work claims that the latent variables, in this case, are able to disentangle content effectively, which I am inclined to agree with, however, I feel that the experimental setup is lacking to solidly support this hypothesis (which I detail in the “cons” and “questions” section).

**Summary Of The Review:**

Overall, although I like intuition and motivation, however, I feel the empirical support is lacking. I would be willing to raise my score if experiments on other aspects or in other languages were done. I believe this would further reinforce the claims made in the paper regarding unsupervised disentanglement.

---
Post rebuttal:

I appreciate the additional experiments the authors provided and their time and effort in writing the rebuttal. This has clarified some of my concerns and added extra evidence to the paper. For this reason, I will raise my score to a 5. Despite this, I still feel that a more general approach, able to disentangle different factors, or in different languages would add more substance that I feel this paper is lacking.

---

> ### Author Response · Authors · 2021-11-15
> **Response to Reviewer WuPD (1/2)**
>
> We thank reviewer WuPD for their comments. Here is an answer to the various points you raised:
>
> ## Cons:
> - **The baseline VAE is not a proper comparison**: We added a Transformer-based baseline with the same Hyper-parameter settings as ADVAE. ADVAE performs better than this baseline, and this shows that the cross-attention in ADVAE is the specific component responsible for its performance.
> - **Reorganizing section 3**: In an effort to follow your recommendation, we made it so that figure 2 and the inference/generation model descriptions are on the same page. That way they can work as a step-by-step guide through the figure, and perhaps make it easier.
> the impact of increasing/decreasing $N_Z$: We added An appendix K with results for $N_Z \in$ {2, 4, 8, 16}. Since the dataset consists of sentences that are 8.92 ± 2.66 tokens long, the $N_Z$ =16 run should have more latent variables than most sentences in the dataset. We could confirm the trend we observed with only 4 and 8: setting $N_Z$ to more than the factors we aim to disentangle (4 syntactic roles) splits the influence on a single syntactic role between more latent variables. Nevertheless, the specialization of latent variables in syntactic roles remains very visible (see the heatmaps we included). This specialization happens, however, in *groups* of latent variables.
> - **Testing how well different aspects (not just syntactic roles) are disentangled**: We also think that studying the disentanglement capabilities of ADVAE with regard to other attributes (such as sentiment), would strengthen our case. However, we prioritized the realizations of core syntactic roles since the motivation of our work revolves around them. And as you can see, prioritizing syntactic roles didn’t leave room for other attributes to be investigated in our paper. Therefore, we delay this investigation to future work. For a study on unsupervised disentanglement of sentences that doesn’t claim to be specific to certain factors, we refer you to the work of Xu et al. (2020) cited in our related works.
>
> ## Minor comments (that we couldn’t apply):
> - **Machine Translation/MT Transformer -> Transformer encoder-decoder models**: We actually chose to call MT Transformers (and not Transformer encoders/decoders) because the encoder/decoder terminology is also used for the VAE paradigm in our work. A previous version of our paper used encoder/decoder for both cases and proofreaders found it confusing.
> - **p_r (the dependency parser?) is not defined**: p_r is defined in lines 9/10 of “Latent variable influence on decoder” (at the top of page 6 in our revised submission). It is the extracted realization of a syntactic role (as per the procedure described in 5.1 “Syntactic role extraction”).
>
> ## Questions:
>  - **Compare the variational model to a denoising AE**: This comparison needs an entirely different evaluation protocol since BART is not generative, and does not provide attention values with regard to the different components of a fixed size sentence representation. However, if the goal is to make the model applicable to large pre-trained models, VAEs have been proven compatible with them in Li et al (2020) through the model Optimus. With more computational resources, this could indeed lead to a better ADVAE in future works.
> - **Concerning the use of self-attention when computing the mean and standard deviation vectors**: Yes, you are right. The $e_{z_i}^{enc}$ vectors do not have to exchange information through self-attention. And since the parameters of self-attention are also fixed, the output of its application to the $e_{z_i}^{enc}$ vectors should also yield fixed vectors that would only use cross-attention. Consequently, the system should behave similarly with or without self-attention. Nonetheless, we are currently rerunning the experiments without that self-attention module to ensure that the results are the same. We will report these results to you as soon as we have them.
>  - **The emergence of our disentanglement in machine translation**: That is indeed an interesting question we would like to investigate in future works. We also think that the same behavior should be present when using different source and target languages.

---

> > ### Author Response · Authors · 2021-11-15
> > **Response to Reviewer WuPD (2/2)**
> >
> >
> > We kindly invite you to refer to the general response to reviewers for a summary of additional results and changes that we brought to the paper. We agree with the reviewer that including many languages in our study would have been interesting, especially for languages with more free word order and richer morphology that could be challenging for our approach and could necessitate much more latent variables to model the interface between morphology and syntax as well as a disentanglement at the character level (cf. case marked in suffix in some languages). We choose instead to first focus on a language heavily used by the community(Xu et al. 2020; Huang & Chang, 2021), first for comparability reasons and then because of the large availability of different data sets with sufficient size. For instance, XNLI(Conneau et al., 2018), the extension of SNLI to 15 languages only consists of 15K sentences (dev+test) and is only made for evaluation.
> >
> > Please feel free to request any other detail you might need.
> >
> > Conneau  et al., (2018)https://arxiv.org/abs/1809.05053
> >
> > Li et al., (2020): https://arxiv.org/pdf/2004.04092.pdf
> >
> > Xu et al., (2020): https://arxiv.org/abs/1905.11975
> >
> > Huang & Chang, (2021): https://aclanthology.org/2021.eacl-main.88/
> >
> > Dittadi et al., (2021): https://openreview.net/forum?id=8VXvj1QNRl1

---

> > > ### Author Response · Authors · 2021-11-17
> > > **Results without self attention on  $e_{z_i}^{enc}$**
> > >
> > > This same response has been addressed to reviewer 9pDc as he asked a similar question:
> > >
> > > We report in the table below the comparison between disentanglement results reported for ADVAE in Table 1, and the same runs for versions of ADVAE that do not use Self-Attention on  $e_{z_i}^{enc}$ (marked with No SA). Similarly, we ran the experiments 5 times and report the numbers in each cell as follows: <mean>(<standard_deviation>).
> > >
> > > | Model | $\beta$ | $\mathbb{D}_{enc}\uparrow$ | $N_{\Gamma^{enc}}\uparrow$ |  $\mathbb{D}_{dec}\uparrow$ | $N_{\Gamma^{dec}}\uparrow$  |
> > > | :---------------: |:----:|:----:|:----:|:----:|:----:|
> > > | Ours-4  |  0.3 | 1.30(0.09) | 3.00(0.00) | 0.68(0.22) | 2.80(0.45) |
> > > | Ours-4, No SA  |  0.3 | 1.48(0.15) | 3.00(0.00) | 0.71(0.06) | 3.00(0.00) |
> > > | Ours-4  |  0.4 | 1.46(0.33) | 3.00(0.00) | 0.81(0.05) | 3.00(0.00) |
> > > | Ours-4, No SA  |  0.4 | 1.43(0.79) | 3.00(0.00) | 0.72(0.37) | 2.80(0.45) |
> > > | Ours-8  |  0.3 | 1.36(0.13) | 3.40(0.89) | 0.60(0.10) | 3.00(0.00) |
> > > | Ours-8, No SA  |  0.3 | 1.34(0.18) | 3.80(0.45) | 0.51(0.14) | 2.80(0.45) |
> > > | Ours-8  |  0.4 | 1.44(0.79) | 3.40(0.55) | 0.63(0.35) | 2.80(0.45) |
> > > | Ours-8, No SA  |  0.4 | 1.75(0.47) | 2.80(0.45) | 0.98(0.27) | 2.60(0.89) |
> > >
> > > As can be seen, all the results without Self-Attention overlap (considering their standard deviation) with the results for ADVAEs that use Self-Attention. Therefore, we cannot conclude that the results are different. We thank you for asking us to clarify this point.
> > >
> > > **Update**:
> > > In the latest rebuttal revision of our submission, we remove the Self-Attention (SA) from the encoder in Figure 2.b, and we replace the results in Table 1 with the results of the model without SA (lines marked with No SA in the table hereabove) since it performs similarly. Due to time constraints, the experiments in the Appendices will be rerun without SA in the encoder for the final version of the paper if it is accepted. This shouldn't pose a problem since *i)* The Appendices mostly show negative results *ii)* The quantitative results in the main body of the paper show similar behavior for ADVAE with or without SA.

---

### Official Review · Reviewer_9pDc · 2021-11-02

**Correctness:** 4
**Technical Novelty And Significance:** 2
**Empirical Novelty And Significance:** 3
**Recommendation:** 8
**Confidence:** 3

**Main Review:**

Strengths:

* The paper studies a research question that is at the heart of representation learning, which could potentially be very impactful
* The model is based on an intuitive and simple idea
* The results are convincing and well presented

Weaknesses:

* No ablation studies. In my opinion, the results are not actually conclusive about what model components are responsible for the improvements. For example, the VAE baseline is based on LSTM encoder-decoder models, whereas the proposed model employs Transformers. Moreover, the VAE baseline seems to be significantly smaller than the Tranformer. Are these two models really comparable enough to conclude that the improvements are due to the product of Gaussians formulation? I think a fairer comparison with minimal changes between the baseline and proposed model could be made as follows: Pool the outputs of 2b) before estimating a single Gaussian distribution, which can then be used as a standard VAE.

* The concrete architecture is not well motivated: It is not clear to me why the inference network architecture has to look the way it does in 2b). Does the model actually use co-attention (i.e., sentence representations and latent variables serve as queries for each other)? Figure 2b) looks like a vanilla Transformer encoder-decoder architecture to me, which uses cross-attention, not co-attention. Moreover, it is not clear that the model actually needs self-attention to refine the latent variables. Wouldn't the model still work if the learnable $e_{z_i}^{enc}$ functioned merely as queries to a single additional attention function on top of the Transformer encoder (followed by a linear layer and softplus)?

* Some related work is not discussed. Specifically, Behjati and Henderson (2021) propose a very similar architecture for learning meaningful units in text: They train an autoencoder with a fixed number of latent vectors via slot attention, which are supposed to capture morphemes. Could their model be applied to disentangle semantic roles, too? In general, how does your work compare to slot attention (Locatello et al. (2020))

Minor:
* Table captions should appear above the table according to the style guide

Locatello et al. (2020): https://arxiv.org/abs/2006.15055

Behjati and Henderson (2021): https://arxiv.org/pdf/2102.01223.pdf

**Summary Of The Paper:**

The paper proposes a method for unsupervised disentanglement of text components and shows its ability to identify semantic roles.
To this end, a neural network is trained to compress the input into a fixed number of independent latent variables which are regularized to be standard Gaussians via the VAE framework. The inference network consists of a Transformer encoder-decoder network, where the decoder inputs correspond to the latent variables, which cross-attend to the outputs of the Transformer encoder, i.e., the encoded sentence. The idea behind this architecture is that attention-based seq2seq architectures align source and target sequences with each other.

The model is evaluated on disentanglement of semantic roles. To this end, they investigate how resampling of individual latent variables impacts the semantic roles in the text generated by the decoder, and how syntactic roles are aggregated into latent vectors via attention. They find that their proposed architecture is more successful at disentangling semantic roles into the latent variables than standard VAEs.

**Summary Of The Review:**

While the research presented in this study is very exciting, it is not quite convincing enough yet for me to trust that the results are actually due to the modeling decisions presented. Moreover, a similar work (Behjati and Henderson (2020)) is not discussed, which casts doubt on the novelty of the approach. I therefore recommend to reject the paper in its current state, but I am happy to change my scores upwards if my concerns above can be resolved.

* UPDATE:  I raised my score slightly as a consequence of the rebuttal phase, where authors conducted an ablation study that now more clearly shows that the improvements over standard VAEs are due to multi-vector nature of the representation rather than just because of using Transformers.

---

> ### Author Response · Authors · 2021-11-15
> **Response to Reviewer 9pDc**
>
> We thank Reviewer 9pDc for meticulously reading the paper, for the very helpful feedback they provided, and for the interest they expressed towards our study.
>
> Hereby is an answer to the weaknesses you discussed:
>
> - **No ablation studies**: As you advised, we added results to the experiments section of our paper for a Transformer-based baseline that uses pooling on the encoder side. Interestingly, it associates the syntactic roles to 3 different latent variables (like ADVAE), but with very low concentration scores (meaning that each syntactic role is significantly influenced by more than one latent variable). This confirms that cross-attention plays a crucial role in ADVAE's ability to disentangle.
> - **The model uses cross-attention (not co-attention) and doesn’t need self-attention on $e_{z_i}^{enc}$**: You are right, it is indeed cross-attention and not co-attention (it has been corrected). You are also right in that the fixed $e_{z_i}^{enc}$ vectors do not need to go through self-attention since this self-attention would also yield fixed vectors. Our model should produce similar results with or without this self-attention, but we are currently running the experiments without it to ensure that this is true. We will report the results here as soon as we have them.
> - **Regarding Behjati and Henderson (2021) and (Locatello et al. (2020))**: We were not aware of the connections between our work and unsupervised object discovery, and we thank you for pointing out this interesting link. There are a number of differences between ADVAE and the slot attention module: mainly, the presence of GRUs in slot attention, the fact that ADVAE is generative, and for Locatello et al. (2020) the fact that it randomly samples $e_{z_i}^{enc}$ vectors instead of training fixed ones (which is necessary for our work). We now include Behjat & Henderson (2021), and Locatello et al (2020) in our Related works as closest to what we do, and we added to our model’s description the fact that using cross attention to obtain representations was first introduced by Locatello et al (2020). Basing our work off minimally modified MT Transformer (instead of slot attention) remains a reasonable choice as it is motivated by an observation on MT transformers. Regarding the model from Behjati &Henderson  (2021), we note that it is not generative, so it couldn’t have been used in conjunction with our evaluation procedure. We would also like to point out that the Arxiv version of our paper predates that Arxiv paper by a little more than a  month.
>
> We hope the above answered your questions, and we kindly invite you to ask further questions otherwise.
> For a summary of the new results and changes applied to our paper, please refer to the general response to reviewers.

---

> > ### Author Response · Authors · 2021-11-17
> > **Results without self attention  on $e_{z_i}^{enc}$**
> >
> > This same response has been addressed to reviewer WuPD as he asked a similar question:
> >
> > We report in the table below the comparison between disentanglement results reported for ADVAE in Table 1, and the same runs for versions of ADVAE that do not use Self-Attention on  $e_{z_i}^{enc}$ (marked with No SA). Similarly, we ran the experiments 5 times and report the numbers in each cell as follows: <mean>(<standard_deviation>).
> >
> > | Model | $\beta$ | $\mathbb{D}_{enc}\uparrow$ | $N_{\Gamma^{enc}}\uparrow$ |  $\mathbb{D}_{dec}\uparrow$ | $N_{\Gamma^{dec}}\uparrow$  |
> > | :---------------: |:----:|:----:|:----:|:----:|:----:|
> > | Ours-4  |  0.3 | 1.30(0.09) | 3.00(0.00) | 0.68(0.22) | 2.80(0.45) |
> > | Ours-4, No SA  |  0.3 | 1.48(0.15) | 3.00(0.00) | 0.71(0.06) | 3.00(0.00) |
> > | Ours-4  |  0.4 | 1.46(0.33) | 3.00(0.00) | 0.81(0.05) | 3.00(0.00) |
> > | Ours-4, No SA  |  0.4 | 1.43(0.79) | 3.00(0.00) | 0.72(0.37) | 2.80(0.45) |
> > | Ours-8  |  0.3 | 1.36(0.13) | 3.40(0.89) | 0.60(0.10) | 3.00(0.00) |
> > | Ours-8, No SA  |  0.3 | 1.34(0.18) | 3.80(0.45) | 0.51(0.14) | 2.80(0.45) |
> > | Ours-8  |  0.4 | 1.44(0.79) | 3.40(0.55) | 0.63(0.35) | 2.80(0.45) |
> > | Ours-8, No SA  |  0.4 | 1.75(0.47) | 2.80(0.45) | 0.98(0.27) | 2.60(0.89) |
> >
> > As can be seen, all the results without Self-Attention overlap (considering their standard deviation) with the results for ADVAEs that use Self-Attention. Therefore, we cannot conclude that the results are different. We thank you for asking us to clarify this point.

---

> > > ### Comment · Reviewer_9pDc · 2021-11-21
> > > **Response to author response**
> > >
> > > Thank you for updating your paper, conducting additional experiments, and discussing Locatello et al. and Behjati and Henderson. I think the inclusion of the Transformer VAE baseline make the paper more convincing. Scientifically, the paper is good enough now for me, so I raised my scores slightly. However, I still have issues that prevent a higher score:
> > > * You conducted experiments with a model whose encoder part doesn't use self-attention on the $e^{enc}$ vectors anymore. If I understand correctly, that model performs just as well as the more complicated model that does use self-attention. But you still chose to go with the more complicated model in the paper. Why? Wouldn't Occam's Razor demand to get rid of unnecessary complexity?
> > > * You cite Locatello et al. (2020) as __"Producing representations in such a manner (with Cross-Attention) has been introduced by Locatello
> > > et al. (2020) as part of the Slot Attention modules in the context of unsupervised object discovery, which is, in computer vision, analogous to what we aim to perform in this work."__ , i.e., a method that relies on cross-attention like your model. I don't think this is a fair representation of your nor their paper - in fact, it doesn't use classical attention at all, which normalizes the attention weights over the keys per query, but rather normalizes them over the queries. making sure that the sum of attention weights a key obtains from any query is one. I believe your model is substantially simpler (a good thing!), but in this quote from your paper you make the impression that ADVAE is the same but for text. Ideally, you could show experimentally that your way of aggregating encoder vectors into target vectors is the better one, or at least point out the advantages of your approach (e.g., being much faster).

---

> > > > ### Author Response · Authors · 2021-11-22
> > > > **About the remaining concerns of Reviewer 9pDc**
> > > >
> > > > We are glad that the changes we brought to our submission do address some of your concerns, and we thank you for providing further pointers for us to improve our paper. Here is our answer to your remaining concerns:
> > > > - **About reporting the results of the model without Self-Attention(SA) in the paper**: You are right, Occam's Razor would demand that we replace the model with the one without SA. We didn't do the replacement in our initial resubmission because we wouldn't be able to obtain the results for all the experiments in the Appendices in time for this first stage of this discussion. However,  the experiments in the appendices are mostly about negative results (Yelp, Hierarchical ADVAE ...),  so reproducing them without SA shouldn't pose a problem. Therefore, in this latest update of our submission, we replace ADVAE in the main body of the paper with the version without SA. This change concerns Figure 2.b and the results in Table 1, along with minor text changes. We will relaunch the experiments in the Appendices, so as to be able to replace them for the final version of the paper if it is accepted.
> > > > - **About the comparison with Locatello et al., (2020)**: We changed the lines you cite in our paper as follows so as to describe more faithfully the differences between ADVAE and Slot Attention:
> > > > "Producing representations with Cross-Attention has been introduced by Locatello et al., (2020) as part of the Slot Attention modules in the context of unsupervised object discovery. However, in contrast to Locatello et al., (2020), we simply use Cross-Attention as it is found in Vaswani et al., (2017), i.e. without normalizing attention weights over the query axis, or using GRUs(Cho et al., 2014) to update representations. As will be shown through our experiments, this is sufficient to disentangle syntactic roles.". The contrast between Slot Attention and ADVAE was indeed oversimplified, we thank you for pointing this out.

---

> > > > > ### Comment · Reviewer_9pDc · 2021-11-22
> > > > > **Thanks**
> > > > >
> > > > > With all the changes you have done, the paper is considerably stronger than the initial submission. I raised my score further. I think this is the kind of paper the ICLR community would be interested in.

---

### Official Review · Reviewer_7uFL · 2021-11-03

**Correctness:** 3
**Technical Novelty And Significance:** 2
**Empirical Novelty And Significance:** 3
**Recommendation:** 5
**Confidence:** 3

**Main Review:**


## Strengths (Reasons to accept)
- The paper is generally well-formed.
- It proposes a new protocol that can be utilized for estimating the extent to which each latent variable is mapped to a specific syntactic role.

## Weaknesses (Reasons to reject)

- I'm not sure this work is the first that attempts to combine Transformers into the VAE framework. For instance, Li et al. (https://arxiv.org/pdf/2004.04092.pdf) already proposed a model consisting of pre-trained Transformer encoders and decoders.
Considering that the main novelty of this paper comes from the fact that the authors suggest exchanging the previous RNN to Transformer, it is doubtful this work has a meaningful novelty.
- It is encouraged for the paper to self-contain the specification of the exact Transformer architecture (one with co-attention, Lu et al (2019)) utilized in this work.
- There is no (theoretical) guarantee that the proposed method is specialized for assigning specific syntactic roles to latent variables, except that the method (by relying on the existing $\beta$-VAE) encourages the latent variables to "generally" represent different aspects of sentences. Please explain why the proposed framework should be decent for disentangling "syntactic roles" rather than other linguistic or semantic properties.



**Summary Of The Paper:**

## Summary

- This paper proposes a probabilistic model called Attention-Driven Variational Autoencoder (ADVAE). This model is another instance of $\beta$-VAE whose encoder and encoders are composed of Transformers rather than previous neural architectures such as RNN.
- The authors aim to disentangle the semantics of latent variables according to some syntactic roles (e.g., nouns and verbs) defined by syntax. To achieve this goal, they suggest employing the combination of Transformer and the existing $\beta$-VAE framework which is known to be effective for disentangling the role of each latent variable in VAE.
- Moreover, this work presents a new way of quantifying syntactic disentanglement between latent variables, relying on the information obtained from the attention matrices of the Transformer architecture.
- The experiments show that the proposed method is quantitatively better than the normal VAE, and that swapping the value of a specific latent variable can impact the generation of the target word (decided by the syntactic role of the latent variable we choose).

**Summary Of The Review:**

To summarize, even though this paper demonstrates that Transformers with $\beta$-VAE can be an attractive option for generating latent variables representing some syntactic roles, its novelty is generally limited to the fact that it proposed a new evaluation protocol aimed at estimating the extent to which each latent variable is mapped to a specific syntactic role.
Therefore, my suggestion is a weak reject.

---

> ### Author Response · Authors · 2021-11-15
> **Response to Reviewer 7uFL**
>
> We thank reviewer 7uFL  for their careful reading and assessment. Here are answers to the weaknesses you discussed:
>
> - **About the difference between ADVAE and combining Transformers into the VAE framework**: Combining Transformers and VAEs is indeed not new (we now clarify it at the beginning of the model description). But ADVAE differs from the previous combinations in that it uses cross-attention to obtain latent variables. To better highlight this difference, a baseline that combines VAE and Transformers in a classical way has been added to the experimental section and has been shown to perform below ADVAE for the task at hand. This shows that the cross-attention used in ADVAE to obtain latent variables is critical to unsupervised disentanglement of syntactic roles. Thank you for pointing that out.
> - **About specifying the Transformer Architecture**: The Transformer architecture we use is the same as that introduced by Vaswani et al. (2017). The hyper-parameters we use are specified in Appendix E.
> -**About the specific use of ADVAE to disentangle syntactic roles**: As explained in section “SYNTACTIC ROLES AND DEPENDENCY PARSING”, realizations of syntactic roles consist of tokens that are more dependent on each other than on the rest of the sentence. Together with the observation that Transformers coherently align spans from different languages, we made the hypothesis that an MT Transformer would disentangle them. The purpose of our work was to verify (empirically) this hypothesis. However, our model could very well also disentangle other aspects of the sentence at the same time. We clarify this point in our introduction (first paragraph on page 2 of our revised submission, last sentence),
>
> We would also like to point out that the main purpose of our paper was not to introduce a novel model, but rather to support the following point with evidence: “The representations of sentences can be disentangled without supervision along the core syntactic roles of these sentences”.
>
> For a summary of all the results we added and the changes we brought to our submission, please refer to the general answer to reviewers.
>
> Vaswani et al. (2017): https://arxiv.org/abs/1706.03762

---

### Official Review · Reviewer_wDkZ · 2021-11-03

**Correctness:** 3
**Technical Novelty And Significance:** 3
**Empirical Novelty And Significance:** 3
**Recommendation:** 6
**Confidence:** 3

**Main Review:**

Strengths:
- Learning the syntactic disentanglement in an unsupervised way.
- Providing a series of evaluation protocol aimed at measuring the disentanglement of syntactic roles, which can be used by other work focus on syntactic disentanglement for sentence representations.

Weakness:
- The paper is hard to follow.
- The evaluation protocol only contains intrinsic evaluation, but how these learned latent variables can help the downstream tasks is not clear. For example, whether this can help unsupervised dependency parsing?

Overall, I think the paper provides an interesting perspective for learning syntactic disentanglement. However, I really suggest the authors to reorganize the paper. I cannot understand the main problem settings until I read the paper twice. To help the reader get the background knowledge, the authors can provide some related work in learning disentangled representations at the very beginning of the introduction (basically just remove some part in Section 6). Another issue is Figure 2, the caption does not indicate what is the input and output of the framework, which is illustrated in the following section instead, and the reader need to refer to the next part of the paper to understand the figure. There are some essential information in the appendix that really need to be moved in to the main body, e.g., some analysis of the latent variables and syntactic roles.

Some questions:
- How to determine the value of N_z? Is this related to the number of syntactic roles?
- The syntactic roles used in evaluation protocol are from a dependency parsing instead of gold standard one.  Have you considered testing on texts that have annotated dependency structures?
- What does the predicative structure in Figure 1 do with the syntactic roles discussed in the paper?

**Summary Of The Paper:**

This paper propose a framework to obtain the disentanglement of syntactic roles as latent variables for sentence representations. The model is an attention-driven VAE which maps syntactic roles to separate latent variables using an encoder-decoder framework. In the second part of the paper, the authors introduce an evaluation protocol to quantify disentanglement between latent variables and spans both in the encoder and in the decoder, which includes syntactic role extraction, latent variable influence on decoder, encoder influence on latent variables and disentanglement metrics.

**Summary Of The Review:**

As a final comment, this work does some contribution for learning and evaluating syntactic disentanglement for sentence representations, and will be helpful to the community. However, the writing of the paper should be improved. For detailed comments please refer to the main review.

---

> ### Author Response · Authors · 2021-11-15
> **Response to Reviewer wDkZ**
>
> We thank reviewer wDkZ for their appreciation and rigorous assessment. Here is our answer to the issues you raised:
>
> ## Weaknesses:
> - **“The paper is hard to follow”**: The model description section has been rearranged to make it easier to go through.
> - **“The evaluation protocol only contains intrinsic evaluation, but how these learned latent variables can help the downstream tasks is not clear. For example, whether this can help unsupervised dependency parsing?”**: Our work aims at improving interpretability and control, and we designed intrinsic evaluation for this purpose. For extrinsic evaluation, we are aware that disentanglement has also been observed to improve transfer (see Higgins et al., (2018) for Reinforcement Learning, and Dittadi et al., (2021) for a study on simulated images). The appearance of understandable concepts in representations seems to help transfer in general, and thus the fact that ADVAE exposes understandable syntax-related variables may very well lead to improvement on other NLP tasks. Previous works on unsupervised disentanglement in NLP, such as Xu et al. (2019), also focus on intrinsic evaluation.
>
>
> As per your recommendation, we moved the related works to the beginning of the paper, mentioned the inputs and outputs of the model in the caption of Figure 2, and put this figure on the same page as the descriptions of the inference model and the generative model to make it easier to follow. However, we could not move any content from the appendix to the body of the paper due to space restrictions.
>
> ## Questions:
> - **“How to determine the value of $N_Z$? Is this related to the number of syntactic roles?”**: Yes, we set $N_Z$ to match the number of factors we aim to disentangle. When we raise the value of $N_Z$ higher, each latent variable remains specialized in a syntactic role, but the influence on each syntactic role progressively splits between more than a latent variable. We updated the paper with a new appendix (K) which illustrates this with the disentanglement measures and heatmaps.
> - **“The syntactic roles used in evaluation protocol are from a dependency parsing instead of gold standard one. Have you considered testing on texts that have annotated dependency structures?”**: We are working with sentences of relatively short length (8.92±2.66 words) that are susceptible to being easily parsable. As any remnant errors will be made consistently across the data set, we do believe that using gold trees, although interesting, will not radically change our results. Ideally, the gold data to use should exhibit low variance in terms of syntactic structure, but we do not know of an annotated dataset with such properties.
> - **“What does the predicative structure in Figure 1 do with the syntactic roles discussed in the paper?”**: The syntactic roles we choose to focus on are core syntactic roles, meaning that they are high up in the traditional oblique hierarchy in syntax. The higher syntactic roles in this hierarchy correspond directly to elements of the predicative structure. We include the predicative structure in Figure 1 to show this correspondence. It is important to highlight this because the predicative structure is what decomposes meaning in a sentence.
> We hope the above answers your questions, and we kindly invite you to ask for clarifications or modifications to the paper if any more are needed.
>
> Please refer to the general response for a summary of the results we added and of the changes we applied to the paper.
>
> Higgins et al., (2018): https://arxiv.org/abs/1707.08475
>
> Xu et al., (2020): https://arxiv.org/abs/1905.11975
>
> Dittadi et al., (2021): https://openreview.net/forum?id=8VXvj1QNRl1

---

> ### Comment · Area_Chair_uugn · 2021-12-05
> **Any updated thoughts/overall judgement?**
>
> Hi Reviewer wDkZ, do you have a final overall opinion based on the updated paper and discussion?

---

### Author Response · Authors · 2021-11-15
**General Response to reviewers**

We, again, express our gratitude to all the reviewers for providing very helpful feedback. We are glad for the interest they expressed in our work.

Here is a summary of the improvements brought to the paper thanks to the reviewers’ comments:
- Figure 1 is now on the same page as the generation and inference model descriptions, which should make it easier to follow the explanations about the inference and generation models.
- Related works have been moved to the beginning of the paper to better frame our work.
- An interesting connection between our work and unsupervised object discovery is now highlighted, as both use cross attention to obtain representations.
- As a consequence, we now refer the reader to that work about unsupervised object discovery (Locatello et al., 2020) as a first introduction of the usage of cross-attention in obtaining representations.
- Results using a Transformer-based baseline that doesn’t use cross-attention to yield representations have been added for comparison in the experiments. These results lead to the following conclusion: The disentanglement capabilities of ADVAE are specifically due to the usage of cross-attention to obtain representations, and not merely to the usage of Transformers.
- An appendix displaying results for ADVAE with a number of latent variables $N_Z \in$ {2, 4, 8, 16} has been added to better see the effect of varying $N_Z$. It further confirms the observation made in the main body of the paper: The more $N_Z$ is raised, the higher the chances that a syntactic role is influenced by more than a latent variable. Nevertheless, heatmaps associated with the ADVAE that uses 16 latent variables show that each variable remains highly specialized in a specific syntactic role and that they are therefore grouped by syntactic roles.
- A number of minor corrections suggested by the reviewers.

Regarding the contribution, we stress that the novelty does not lie in the
definition of a VAE based on a seq2seq Transformer instead of an LSTM, but on the
soft mapping from a fixed number of latent variables (LVs) to spans of text. Indeed this
mapping is obtained through Transformer's cross-attention but it may be obtainable by other models that use cross-attention (e.g. using attention based on LSTM hidden states). By
forcing LVs to read spans, disentanglement could be understood as assigning LVs to spans (like a soft version of typing found in programming languages). We then use a parser to understand the relation of LVs to syntactic roles via a "perturb and generate" procedure, and via the values of attention. Through the evidence we obtain, we claim that we can map syntactic realizations with latent variables without supervision, which constitutes our main contribution and the main novelty in our work.

Locatello et al. (2020): https://arxiv.org/abs/2006.15055

---

### Comment · Area_Chair_uugn · 2021-12-05
**Are you really getting grammatical roles?**

While I accept that Yelp is a harder dataset with more variable syntactic constructions, don't the results in Appendix B tend to undermine the idea that in general this method is able to disentangle grammatical roles. E.g., doesn't Figure 7 show that 4 of the 8 latent variables strongly represent the subject while none of the 8 strongly represent the verb, dobj, or pobj (that is, appreciably more than the subject).

---

### Decision · Program_Chairs · 2022-01-20

**Decision:**

Reject

**Comment:**

This paper proposes improving human interpretability and manipulability of neural representations by obtaining syntactic roles (here, subject, object, prepositional object, and main verb) without supervision by means of them becoming linked to latent variables in a novel proposed attention-driven VAE (ADVAE) model, which provides cross attention between a language transformer and latent variables. The paper argues that syntactic roles are quite central to meaning interpretation and that the ADVAE recovers them better than LSTM or Transformer (with mean pooling) VAEs.

This is a quite interesting direction and paper. There was active discussion with the reviewers, one of whom (9pDc) moved their rating from reject to quite strong support, while the other reviewers either sat on the fence or raised from reject to borderline. Nevertheless, I overall tend to agree that the paper is still lacking in empirical support, a view clearly shared by reviewers WuPD and 7uFL. The SNLI data is very simple descriptive sentences, nearly all in the form of S V O or S V PP. Would this work on more complex data, in other languages, or with more word order variation? There isn't very much investigation, but the new results added during reviewing based on Yelp data seem to offer more concerns than confidence. These are also very short sentences but with more varied structure and some complementation. It seems like D_{dec} is now very low (much lower than for the sequence VAE), the ability to distinguish out grammatical roles seems limited to {subj} vs. {dobj, pobj} in the encoder and none at all in the decoder (Figure 6/7). And then for the examples in Appendix D, the disentanglement abilities barely seem stronger than being able to pick out subjects, though when there are sentences with subordinate clauses, it is perhaps random which subject you get. The resampled realizations in appendix H also seem to show limited disentanglement: resampling the subject usually seems to change the object as well, often markedly. No convincing downstream applications are shown. As such, while I agree that disentanglement is at the heart of representation learning, I can't get on board with reviewer 9pDc feeling that this paper now has convincing results. Reviewer 7uFL also emphasizes that there is no strong reason that the latent variables have to align with syntactic roles. In particular, the motivation in NMT whereby constituents clump and reorder together does not exist here. It may only work for the very simple and regular sentences of SNLI.

Hence, overall, I feel that this method needs more extensive validation on harder, more varied data sets before it becomes a convincing contribution, and so I propose rejecting the paper at this point in time. Nevertheless, I do think the topic is interesting and this approach has the potential to be good.